# Effective binning of metagenomic contigs using contrastive multi-view representation learning

Ziye Wang [1], Ronghui You[1], Haitao Han [1], Wei Liu[1], Fengzhu Sun [2] & Shanfeng Zhu [1,3,4,5,6] ✉

Contig binning plays a crucial role in metagenomic data analysis by grouping contigs from the same or closely related genomes. However, existing binning methods face challenges in practical applications due to the diversity of data types and the difficulties in efficiently integrating heterogeneous information. Here, we introduce COMEBin, a binning method based on contrastive multi-view representation learning. COMEBin utilizes data augmentation to generate multiple fragments (views) of each contig and obtains high-quality embeddings of heterogeneous features (sequence coverage and k-mer distribution) through contrastive learning. Experimental results on multiple simulated and real datasets demonstrate that COMEBin outperforms state-of-the-art binning methods, particularly in recovering near-complete genomes from real environmental samples. COMEBin outperforms other binning methods remarkably when integrated into metagenomic analysis pipelines, including the recovery of potentially pathogenic antibiotic-resistant bacteria (PARB) and moderate or higher quality bins containing potential biosynthetic gene clusters (BGCs).

Metagenomics provides a culture-free method to study microorganisms by directly analyzing genomes and genes collected from microbial populations[1] and mining important information related to the environment and human diseases[2–4]. Corresponding computational methods have been developed rapidly in the past two decades, including metagenome assembly, contig binning, and microbial profiling[5]. Binning, which groups DNA fragments (such as contigs) from the same or close genome into the same bin, is important in analyzing metagenomic sequencing data. The quality of the binning results will affect the microbial community structure and function analysis, the discovery of microbes, and other follow-up metagenomics investigations[2,6,7]. How to provide reliable and high-performance contig binning results for metagenomics research has become an important research topic.

Many binning methods have been developed. According to the features utilized, they can be mainly divided into three groups: sequence composition (k-mer frequency) based, abundance (contig coverage) based, and hybrid methods (combine both the k-mer frequency and coverage features)[8]. Although hybrid methods are usually superior to those using only one kind of information, efficiently integrating heterogeneous features remains challenging. Researchers have made many attempts to address this problem. For example, CONCOCT[9] directly concatenates two kinds of features, easily affected by the normalization method of heterogeneous features and the number of sequencing samples. MaxBin2[10,11] multiplies the probabilities obtained by the two kinds of features, which requires high accuracy of the two kinds of information. Moreover, MaxBin2 can be computationally prohibitive on datasets with a substantial number of sequencing samples. MetaBAT1[12]

[1]Institute of Science and Technology for Brain-Inspired Intelligence and MOE Frontiers Center for Brain Science, Fudan University, Shanghai, China. [2]Department of Quantitative and Computational Biology, University of Southern California, Los Angeles, CA, USA. [3]Shanghai Qi Zhi Institute, Shanghai, China. [4]Key Laboratory of Computational Neuroscience and Brain-Inspired Intelligence (Fudan University), Ministry of Education, Shanghai, China. [5]Shanghai Key Lab of Intelligent Information Processing and Shanghai Institute of Artificial Intelligence Algorithm, Fudan University, Shanghai, China. [6]Zhangjiang Fudan International Innovation Center, Shanghai, China. ✉e-mail: zhusf@fudan.edu.cn

combines two distance measures by weighted summation so that only the linear relationship between the distances of the two kinds of features can be obtained. As an upgraded version, MetaBAT2[13] integrates two distance scores by computing their geometric mean. The recently developed MetaDecoder[14] is a two-layer contig binning model using a modified Dirichlet Gaussian mixture model to create preliminary clusters based on k-mer frequency and coverage. Then, it employs a semi-supervised k-mer frequency probabilistic model and a modified Gaussian mixture model for coverage to generate pure clusters.

The emergence of deep learning-based binning methods has provided improved capabilities in handling heterogeneous information. VAMB[15] connects the two heterogeneous features, including oligonucleotide frequency and coverage features, and obtains latent representations through variational autoencoder[16]. However, VAMB does not use additional information to guide the learning of representations for clustering. CLMB[17] extends the methodology of VAMB by incorporating contrastive learning. Contrastive learning is a self-supervised learning technique used for learning an informative representation of the input data by bringing similar instances closer together while pushing dissimilar instances farther apart[18]. CLMB introduces a pair of augmented data for each contig by adding noise to the feature vector. Subsequently, CLMB obtains the embedded representations that integrates heterogeneous features. Nevertheless, adding simulated noise for data augmentation in CLMB has only slight improvement compared to VAMB due to the difficulty in simulating the data noise for the complex metagenomes. SemiBin1[19] is a semi-supervised binning algorithm based on deep learning. It constructs pairwise must-link constraints by splitting long contigs into two equal-length segments and constructs pairwise cannot-link constraints from the taxonomic annotation information. SemiBin1 utilizes a semi-supervised autoencoder to extract the constraint information and obtain embeddings for subsequent clustering. As an upgraded version, SemiBin2[20] adopts the same approach as SemiBin1 for generating must-link constraints, but it introduces cannot-link constraints through random sampling of pairs of contigs. However, the quantity and quality of the must-link constraints are influenced by the length distribution of the contigs within the datasets.

Here we propose COMEBin, a contig binning method based on contrastive multi-view representation learning. The key contributions of COMEBin can be summarized as follows: 1) We introduce a data augmentation approach that generates multiple views for each contig, enabling contrastive learning and yielding high-quality representations of the heterogeneous features; 2) COMEBin incorporates a "Coverage module" to obtain fixed-dimensional coverage embeddings, which enhances its performance across datasets with varying numbers of sequencing samples; 3) COMEBin employs the advanced community detection algorithm, Leiden[21], for clustering. Moreover, we adapt the settings of Leiden specifically for the binning task, considering single-copy gene information[22] and contig length. This adaptation ensures that COMEBin produces robust and reliable binning results across diverse datasets.

Recently, three kinds of binning modes have been used in related studies[5,15,19,23]: co-assembly, single-sample, and multi-sample binning. In single-sample binning, each sequencing sample is independently assembled and binned. In multi-sample binning, each sequencing sample is still individually assembled, but the sequencing reads from all the samples are used for generating the abundance for binning. In contrast to single- and multi-sample binning, different sequencing samples are pooled together for assembling and binning in co-assembly binning. We have validated the performance of COMEBin on ten simulated and six real datasets in co-assembly binning and three real datasets in single- and multi-sample binning. Advanced binning methods for comparison include three widely used binning methods (CONCOCT[9], MetaBAT2[12,13], and MaxBin2[10,11]), four deep learning-based binning methods (VAMB[15], CLMB[17], SemiBin1[19], SemiBin2[20]), and a newly published binner, MetaDecoder[14].

On the sixteen datasets evaluated in co-assembly binning, COMEBin achieves the best overall performance. For example, COMEBin achieves the best results regarding the number of recovered near-complete bins (>90% completeness and <5% contamination) on fourteen datasets. Compared with the best of other methods (the best results of other binners for all datasets come from different binning methods), COMEBin has an average improvement of 9.3% and 22.4% on the simulated datasets and the real datasets, respectively. In the evaluation conducted on the three datasets in both single- and multi-sample binning, COMEBin has an average improvement of 33.2% and 28.0%, respectively, compared with the best of other methods.

We replaced the advanced binning methods with COMEBin in metagenomic analysis pipelines, including identifying potential pathogenic antibiotic-resistant bacteria (PARB) and recovering moderate or higher quality bins containing potential biosynthetic gene clusters (BGCs). COMEBin increases the number of identified potential PARB by an average of 33.3%, 74.5%, and 60.5% in comparison to the utilization of MetaBAT2, MetaDecoder, and SemiBin2, respectively. COMEBin recovers 126% and 70.6% more moderate or higher quality bins containing at least one BGC, compared to the second-best performers in single- and multi-sample binning, respectively.

## Results
### COMEBin outperforms other binning methods on simulated datasets

To compare COMEBin with other binning methods on the simulated datasets, we used ten benchmark datasets, including four CAMI II toy datasets (CAMI Gt, CAMI Airways, CAMI Skin, and CAMI Mouse gut) and six benchmark datasets from the second round of CAMI challenges[5] (Marine GSA, Marine MA, Plant-associated GSA, Plant-associated MA, Strain-madness GSA, and Strain-madness MA), where "GSA" denotes gold standard assembly and "MA" denotes MEGAHIT assembly. See the "Methods" section for details.

For the four CAMI II toy datasets, COMEBin outperforms other binning methods, including deep learning-based binning methods (VAMB, CLMB, SemiBin1, and SemiBin2), in terms of the number of recovered near-complete bins (>90% completeness and <5% contamination; see Fig. 1a). Notably, COMEBin achieves these results without relying on semi-supervised taxonomic information used by SemiBin1. Compared to the second-best methods, COMEBin increases the number of recovered near-complete bins from 135, 135, 154, and 415 to 156, 155, 200, and 516, respectively. Figure 1b illustrates that COMEBin consistently attains the highest accuracy values across all the datasets. Moreover, as shown in Supplementary Fig. S1, COMEBin achieves the best overall performance in terms of the F1-score (bp), Adjusted Rand Index (bp), percentage of binned bp, and accuracy (bp) metrics. The notation "(bp)" indicates that the evaluations are based on base pair counts as done in refs. 5,23,24.

For the CAMI II challenge datasets, COMEBin performs best on the four marine and plant-associated datasets, as shown in Fig. 2a, and the second-best methods are different. Especially on the Marine GSA dataset, COMEBin increases the number of recovered near-complete bins from 285 to 337 compared with the second-best method. The quality of assemblies has a significant impact on binning performance. All the binners perform better on the gold standard assembly (GSA) than the corresponding MEGAHIT assembly (MA), and the average number of recovered near-complete genomes of Marine, Plant-associated, and Strain-madness datasets have increased by 218%, 242%, and 318%, respectively, when transitioning from MA to GSA datasets. MaxBin2, SemiBin1, and SemiBin2 are particularly influenced by assembly quality, potentially due to the utilization of single-copy gene information in clustering. Supplementary Fig. S2 shows the performance of the F1-score (bp), ARI (bp), percentage of binned bp, and accuracy (bp). In terms of the accuracy values on different datasets (see Fig. 2b and Supplementary Fig. S2), COMEBin achieves the best performance.

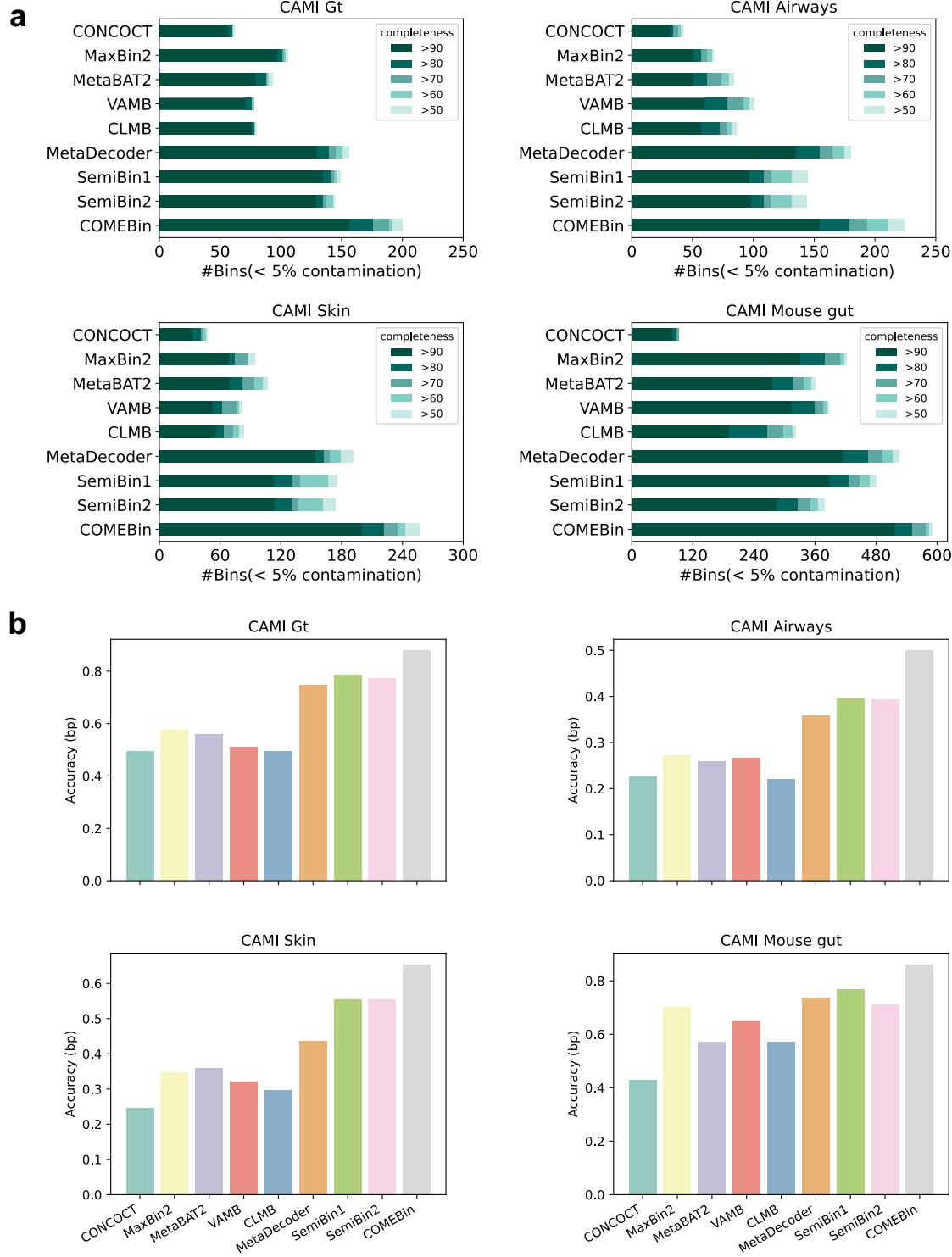

**Fig. 1 | Comparison of binning methods on the four simulated datasets. a** The number of recovered bins with contamination <5% and varying completeness thresholds. **b** The accuracy (bp) for the binning methods.

Regarding the strain-madness datasets, no single method exhibits a distinct advantage. The presence of closely related strains (genomes with an average nucleotide identity (ANI) of no less than 95% to at least another genome) in the strain-madness datasets poses challenges for recovering high-quality bins using existing binning methods. COMEBin (no k-mer) is a variant of COMEBin that only utilizes coverage information. It recovers 52 near-complete bins on the Strain-madness GSA dataset, surpassing COMEBin, which recovers 39 near-complete bins.

Additionally, all methods perform poorly on the Strain-madness MA, indicating the continued difficulties in assembly and binning for datasets containing highly similar strains.

**COMEBin outperforms other binning methods on real datasets**
We utilized CheckM2[25] to obtain the numbers of the high-quality bins recovered by different binning methods. Figure 3a illustrates that COMEBin consistently outperforms other binners across all six datasets.

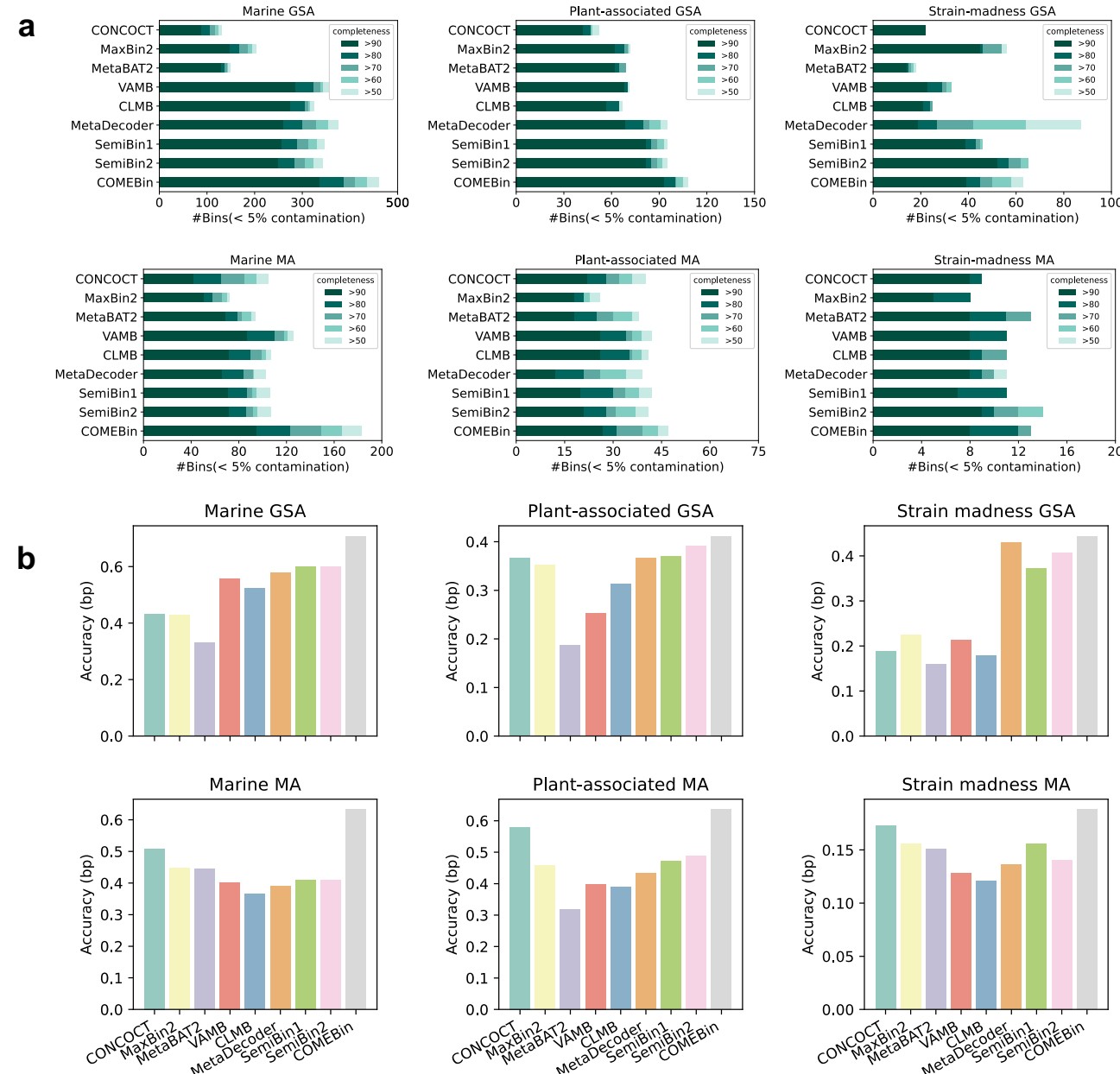

**Fig. 2 | Comparison of binning methods on CAMI II datasets. a** The number of recovered bins with contamination <5% and varying completeness thresholds. **b** The accuracy (bp) for the binning methods.

While VAMB performs well on the two datasets with a higher number of samples (STEC and MetaHIT datasets), its effectiveness diminishes on the three Water Group datasets, which consist of fewer than ten sequencing samples. One possible explanation is that VAMB directly concatenates two types of features and learns embeddings of heterogeneous data, making it susceptible to the dimensionality of coverage features (i.e., the number of sequencing samples). Conversely, COMEBin still achieves good performance on the datasets with few samples. Firstly, COMEBin incorporates a "Coverage network" module, capable of generating fixed-dimensional embedded representations for coverage features. Furthermore, COMEBin employs contrastive learning, enabling the discovery of more informative features.

In terms of the number of near-complete genomes recovered by the binning methods, COMEBin is the best-performing method. VAMB is the second-best method on the datasets with more than 50 sequencing samples (STEC and MetaHIT datasets), while MetaBAT2 is the second-best performer on the three Water Group datasets with less

than ten sequencing samples. On average, COMEBin outperforms the second-best performing methods by recovering 22.4% more near-complete bins. It is worth highlighting that the second-best results are achieved by different binners across the datasets.

We further annotated the bins with >50% completeness and <5% contamination produced by MetaBAT2 and COMEBin on the holdout datasets (refer to "Benchmark datasets" section) using GTDB-Tk[26,27]. As illustrated in Fig. 3b, COMEBin recovers more distinct taxa at the various taxonomic levels. Additionally, Supplementary Fig. S3 demonstrates that COMEBin recovers more known and unknown moderate or higher quality bins at the species level in the real datasets.

## COMEBin demonstrates effectiveness across diverse datasets and different binning modes

We summarize the comparison results obtained from the holdout datasets (refer to "Benchmark datasets" section) and demonstrate the usability of COMEBin for single-sample and multi-sample binning on

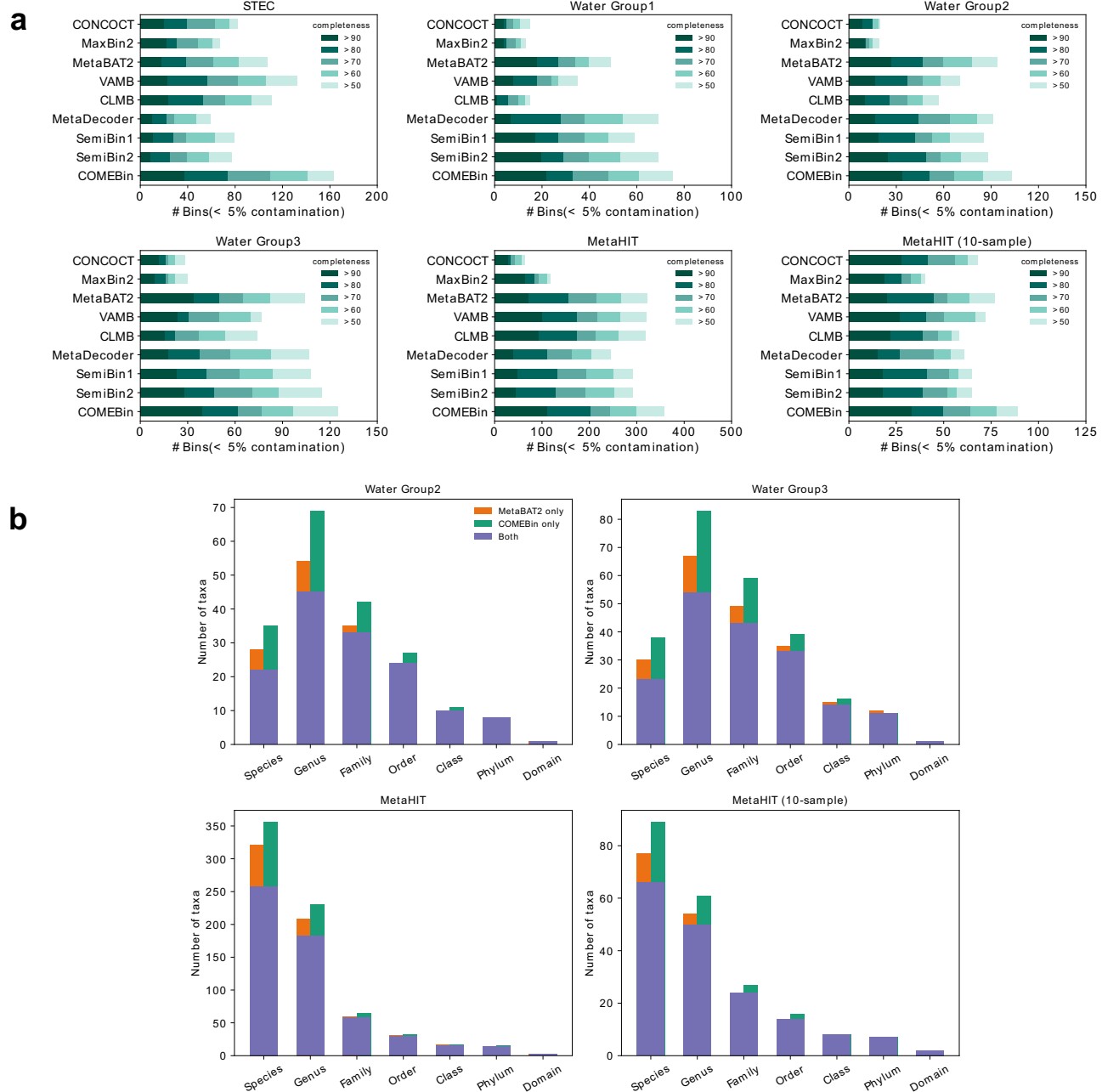

**Fig. 3 | COMEBin outperforms other binners in real datasets. a** The number of recovered bins with contamination <5% and varying completeness thresholds. **b** The numbers of distinct taxa at the species, genus, family, order, class, phylum, and domain levels. GTDB-Tk was utilized to annotate the bins with >50% completeness and <5% contamination.

three real datasets (Water Group2, Water Group3, and MetaHIT (10-sample)). Detailed descriptions of binning modes can be found in the "Binning modes" section.

Regarding the simulated holdout datasets, COMEBin exhibits the best performance regarding the number of bins with an F1-score greater than 0.9, except the Strain-madness GSA dataset, as shown in Supplementary Fig. S4. Notably, COMEBin outperforms the second-highest performers by recovering 19.4% and 19.8% more bins with an F1-score greater than 0.9 in the Marine GSA and Marine MA datasets, respectively. Furthermore, we evaluated the performance of COMEBin on challenging common strains, defined as genomes with an average nucleotide identity (ANI) of no less than 95% to at least another genome in the dataset[5]. Remarkably, COMEBin recovers 16.7% more bins with an F1-score greater than 0.9 compared to the second-highest

performer in the Marine GSA (common) dataset. COMEBin (no k-mer), a variant of COMEBin utilizing only coverage information, recovers 14.9% more bins with an F1-score greater than 0.9 compared to the second-best performer in the Strain-madness (common) dataset, as shown in Supplementary Fig. S5. These results underscore COMEBin's capability to handle highly similar strains effectively.

To further demonstrate the usability of COMEBin across different binning modes, we evaluated COMEBin's performance in single- and multi-sample binning on the MetaHIT (10-sample), Water Group2, and Water Group3 datasets. Supplementary Fig. S6 provides a detailed comparison. COMEBin achieves the best overall performance in both modes. It recovers an average of 33.2% and 28.0% more near-complete genomes compared to the best of other methods in single- and multi-sample binning, respectively.

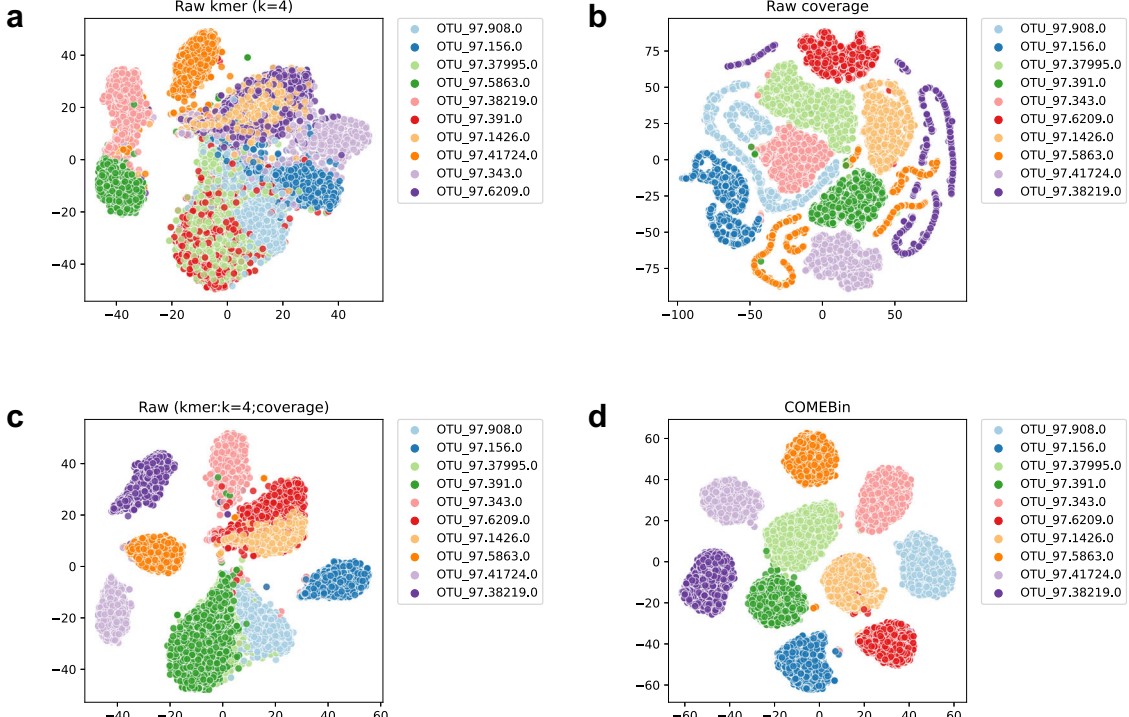

**Fig. 4 | Feature visualization of the CAMI Airways dataset (10 genomes with the highest number of contigs).** The t-SNE[28] was employed to visualize the raw features and the embedded representations derived by COMEBin. Each point represents a contig, and points with the same color indicate that they originate from the same genome. **a** Raw data of *k*-mer frequency features. **b** Raw data of coverage features. **c** Raw data of concatenated features. **d** COMEBin-encoded data.

## Contrastive multi-view learning improves binning results

We ran different variants of COMEBin for comparison on six datasets to show the effect of contrastive multi-view learning. The following describes the variants of COMEBin.

(1) **COMEBin NoContrast (combine)**

The "COMEBin NoContrast (combine)" method uses the concatenated original nucleotide frequency features and coverage features as the inputs and uses the same method of COMEBin to cluster the contigs.

(2) **COMEBin NoContrast (coverage)**

The "COMEBin NoContrast (coverage)" method uses the original coverage features as the inputs and uses the same method of COMEBin to cluster the contigs.

(3) **COMEBin NoContrast (k-mer)**

The "COMEBin NoContrast (k-mer)" method uses the original k-mer frequency features as the inputs and uses the same method of COMEBin to cluster the contigs.

(4) **COMEBin (VAMB embeddings)**

The "COMEBin (VAMB embeddings)" method uses the embeddings generated by VAMB[15] as the inputs of the COMEBin clustering methods.

(5) **COMEBin (CLMB embeddings)**

The "COMEBin (CLMB embeddings)" method uses the embeddings generated by CLMB[17] as the inputs of the COMEBin clustering methods.

(6) **COMEBin (SemiBin1 embeddings)**

The "COMEBin (SemiBin1 embeddings)" method uses the embeddings generated by SemiBin1[19] as the inputs of the COMEBin clustering methods.

(7) **COMEBin (SemiBin2 embeddings)**

The "COMEBin (SemiBin2 embeddings)" method uses the embeddings generated by SemiBin2[20] as the inputs of the COMEBin clustering methods.

As shown in Supplementary Fig. S7, COMEBin achieves much better performance than the "COMEBin NoContrast" methods on all datasets. To further evaluate the effectiveness of COMEBin's embeddings for binning, we utilized the embeddings generated by VAMB, CLMB, SemiBin1, and SemiBin2 as inputs for the COMEBin clustering methods. The results demonstrate that COMEBin's embeddings outperform the alternative approaches, emphasizing their potency for binning. Notably, compared with COMEBin (CLMB embeddings), COMEBin exhibits an average improvement of 50.4% in the number of recovered near-complete genomes. These findings indicate that contrastive multi-view learning enhances the quality of binning results.

We further employed t-SNE[28] to visualize the raw features and the embedded representations derived by COMEBin, which incorporate heterogeneous information. Figure 4 illustrates the feature visualization results of the ten genomes with the largest number of contigs (longer than 1000bp) in the CAMI Airways dataset. Figure 4a presents the visualization of nucleotide frequency features. The results reveal that, except for the three categories at the edges, the contigs of other genomes exhibit mixing with no discernible boundaries. Figure 4b corresponds to the visualization results using only the original contig coverage features, where seven clusters exhibit distinct boundaries. Subsequently, Fig. 4c demonstrates the visualization results after concatenating the two original features. Although there is a significant improvement for the "OTU_97.38219.0" cluster compared to using only the coverage feature, only four clusters exhibit distinct boundaries. Finally, Fig. 4d displays the visualization of the embedded representations with heterogeneous information obtained by COMEBin. The results clearly indicate strong separability among the embedded features of the ten genomes, with distinct boundaries observed between each cluster.

Our method generates five sets of augmented data for each dataset, resulting in six views for each contig, including the original view. To investigate the impact of augmented data volume, we

conducted experiments by varying the number of augmented data sets and evaluated the performance on four simulated and two real datasets. The results, as presented in Supplementary Fig. S8, indicate that the overall performance improves as the number of views increases for 2, 4, and 6 views. Regarding the real datasets, the results obtained using six views exhibit similar performance to those achieved with four and eight views.

## Running time and memory usage

We measured the running time and memory usage for COMEBin, SemiBin2, and VAMB across CAMI Gt and STEC datasets in co-assembly binning, as well as ten Bermuda-Atlantic Time-series Study (BATS) metagenomes in both single- and multi-sample binning (Supplementary Table S1). More details on the BATS samples are given in Supplementary Table S2. The binners were executed on a machine with two 2.50 GHz Intel Xeon Processor E5-2678 CPUs and an RTX 4090 GPU in both CPU-only and GPU modes. We ran the binners with 48 threads. We excluded the running time for aligning reads to contigs, as this step is necessary for all binning methods. We ran each binner on each dataset three times and computed the average running time and memory usage. On the STEC dataset, which includes 53 sequencing samples and over 250,000 contigs, COMEBin's memory usage does not exceed 11GB. Additionally, the GPU version of COMEBin runs for ~6 h. As shown in Supplementary Table S1, the running time of COMEBin varies greatly between CPU-only and GPU modes. The running time of COMEBin in GPU mode is comparable to SemiBin2, but in CPU-only mode, its running time is noticeably longer than the other two methods. We recommend using COMEBin in GPU mode.

## COMEBin assists analysis of potential pathogenic antibiotic-resistant bacteria (PARB)

Identifying potential pathogenic antibiotic-resistant bacteria (PARB) is crucial in microbiological risk assessment[29]. And the three Water Group datasets used for benchmarking in our study were sampled to analyze potential PARB in aquatic environments[29]. To demonstrate the ability of COMEBin to assist the microbiological risk assessment, we compared the results of COMEBin and the advanced binners (MetaBAT2, MetaDecoder, and SemiBin2) in the recovery of potential PARB on the three datasets. Following the study[29], a "moderate or higher" quality bin with at least one antibiotic resistance gene (ARG) and one virulence factor gene (VFG) is defined as a potential PARB.

When integrating COMEBin into the PARB identification workflow, as depicted in Fig. 5a, the number of potential PARB identified from the Water Group datasets has shown an average increase of 33.3%, 74.5%, and 60.5% in comparison to the utilization of MetaBAT2, MetaDecoder, and SemiBin2, respectively (see Fig. 5b). This improvement underscores the effectiveness of COMEBin in enhancing the identification of PARB, thus facilitating more accurate microbiological risk assessment.

## COMEBin helps to recover moderate or higher quality bins carrying potential BGCs

Secondary metabolites in bacteria and fungi are bioactive compounds with potential anti-tumor or antibiotic activities, making them valuable resources for drug discovery[30]. Biosynthetic gene clusters (BGCs) contain the genes responsible for the production of these secondary metabolites[31,32]. Exploring metagenomes for potential BGCs is crucial for unlocking their therapeutic potential. To demonstrate the effectiveness of COMEBin in recovering moderate or higher quality bins containing potential BGCs, we evaluated COMEBin by comparing its performance to MetaBAT2, MetaDecoder, and SemiBin2 using ten metagenomes from the Bermuda-Atlantic Time-series Study (BATS)[33]. Both single-sample and multi-sample binning were evaluated. Binning results were assessed using CheckM2 to identify moderate or

higher quality bins. Potential BGCs were identified in these bins using a secondary metabolite genome mining tool, antiSMASH[34]. COMEBin outperforms the compared binning methods in recovering moderate or higher quality bins in both binning modes (see Fig. 5c). In single-sample binning, it demonstrates a notable 162% enhancement in the number of potential BGCs from the recovered moderate or higher quality bins, compared to the second-best performer (MetaDecoder) (see Fig. 5d). COMEBin recovers 126% and 70.6% more moderate or higher quality bins containing at least one BGC, compared to the second-best performers in single- and multi-sample binning, respectively (refer to Fig. 5e). Moreover, the total lengths of BGCs within moderate or higher quality bins recovered by COMEBin exhibit a 64.8% improvement over the second-best performer in single-sample binning (refer to Fig. 5f). These findings highlight the substantial potential of COMEBin for metagenomic research focused on secondary metabolite discovery.

## Discussion

COMEBin is a binning method based on multi-view contrastive learning. The contrastive learning approach aims to make the representations of different views of the same contig as similar as possible while ensuring that the representations of different contigs are distinct, resulting in highly discriminative embeddings (refer to Fig. 4). In addition, we incorporate the "Coverage network" to obtain fixed-dimensional coverage representations. Compared to VAMB[15], COMEBin demonstrates superior stability in scenarios where the number of sequencing samples varies significantly. Finally, we employ the Leiden-based clustering method to obtain accurate binning results. We optimize the settings of Leiden for the binning task, considering single-copy marker information and contig length. As a result, COMEBin consistently delivers excellent and robust binning results across sixteen simulated and real datasets. Furthermore, to demonstrate COMEBin's performance on low-complexity datasets, we randomly selected ten genomes from the CAMI Skin and CAMI Mouse gut datasets for testing. As shown in Supplementary Fig. S9, COMEBin consistently recovers the most near-complete genomes.

CLMB[17] is an existing contrastive learning-based binning method, which constructs a pair of augmented data for each contig for contrastive learning by adding noise to the feature vectors. However, simulating the data noise for the complex metagenomes is challenging, affecting the quality of the augmented data and resulting in limited improvements compared to its non-contrastive version VAMB[15]. In contrast to CLMB, COMEBin employs a more effective data augmentation approach and generates multiple views for each sequence by randomly extracting multiple continuous fragments from each contig. The multiple-view strategy improves binning performance (see Supplementary Figs. S7 and S8). Especially, COMEBin outperforms CLMB in recovering near-complete genomes, exhibiting an average improvement of 66.1% across ten holdout datasets in co-assembly binning.

COMEBin demonstrates effectiveness across different binning modes. In addition to evaluating its performance on the benchmark datasets in co-assembly binning, we demonstrate the utility of COMEBin on the MetaHIT (10-sample), Water Group2, and Water Group3 datasets in single- and multi-sample binning. COMEBin recovers an average of 33.2% and 28.0% more near-complete genomes compared to the best of other methods in single- and multi-sample binning, respectively (see Supplementary Fig. S6).

We have also compared COMEBin with SemiBin2, MetaDecoder, and MetaBAT2 using four long-read sequencing datasets, as stated in the Supplementary Note. Long-read sequencing typically yields highly contiguous assemblies, resulting in fewer contigs and smaller bins (measured by the number of contigs)[20]. According to the results shown in Supplementary Fig. S10, SemiBin2 (long-read mode) performs best, followed by COMEBin. In future research, we plan to design a binning

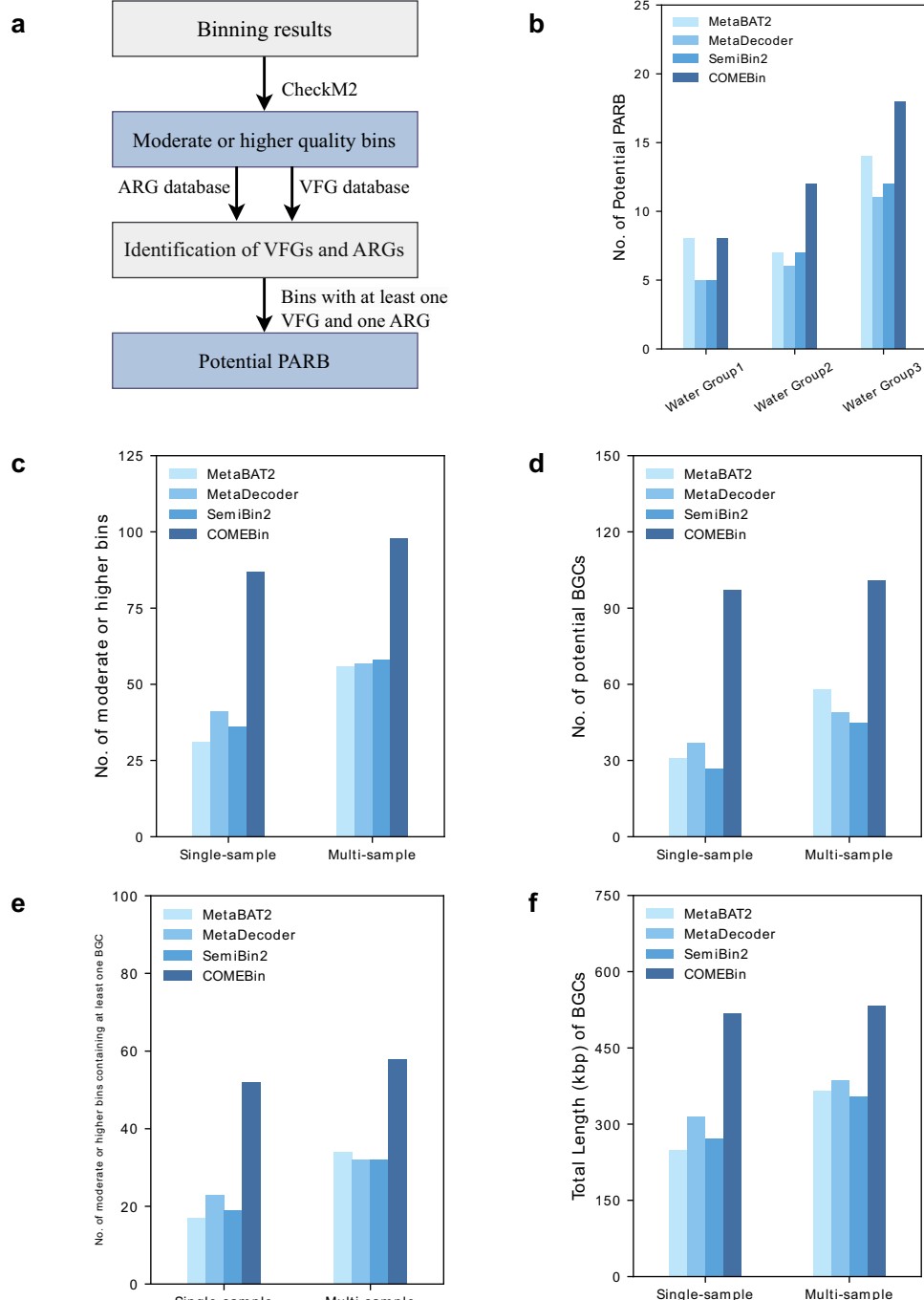

**Fig. 5 | Recover the potential pathogenic antibiotic-resistant bacteria (PARB) and moderate or higher quality bins containing potential BGCs. a** Pipeline for recovering the potential PARB. After obtaining the binning results, we evaluated the bin quality using CheckM2, identifying moderate or higher quality bins (>50% completeness and <10% contamination). Subsequently, Resistance Gene Identifier[46] and BLASTP[47] were utilized to predict antibiotic resistance genes (ARGs) and virulence factor genes (VFGs) within these bins. Moderate or higher quality bins containing at least one VFG and ARG were categorized as PARB. **b** The number of potential PARB identified in Water Group datasets. **c** The number of moderate or higher quality bins in ten BATS samples (sum) within single- and multi-sample binning. **d** The number of potential BGCs in moderate or higher quality bins in ten BATS samples (sum) within single- and multi-sample binning. **e** The number of moderate or higher quality bins containing at least one BGC in ten BATS samples (sum) within single- and multi-sample binning. **f** Total Length (kbp) of BGCs identified in moderate or higher quality bins in ten BATS samples (sum) within single- and multi-sample binning.

algorithm specifically for long-read sequencing data and explore the application of the multi-view contrastive learning method on third-generation sequencing reads, eliminating the need for generating assemblies.

In summary, COMEBin outperforms existing state-of-the-art binning methods in recovering individual genomes from complex microbial communities, as demonstrated by extensive experiments. Furthermore, COMEBin can be a valuable tool for analyzing metagenomic sequencing data, and we encourage researchers to integrate it into their metagenomic analysis pipelines, including identifying pathogenic antibiotic-resistant bacteria and potential biosynthetic gene clusters (BGCs) in metagenomes (See Fig. 5).

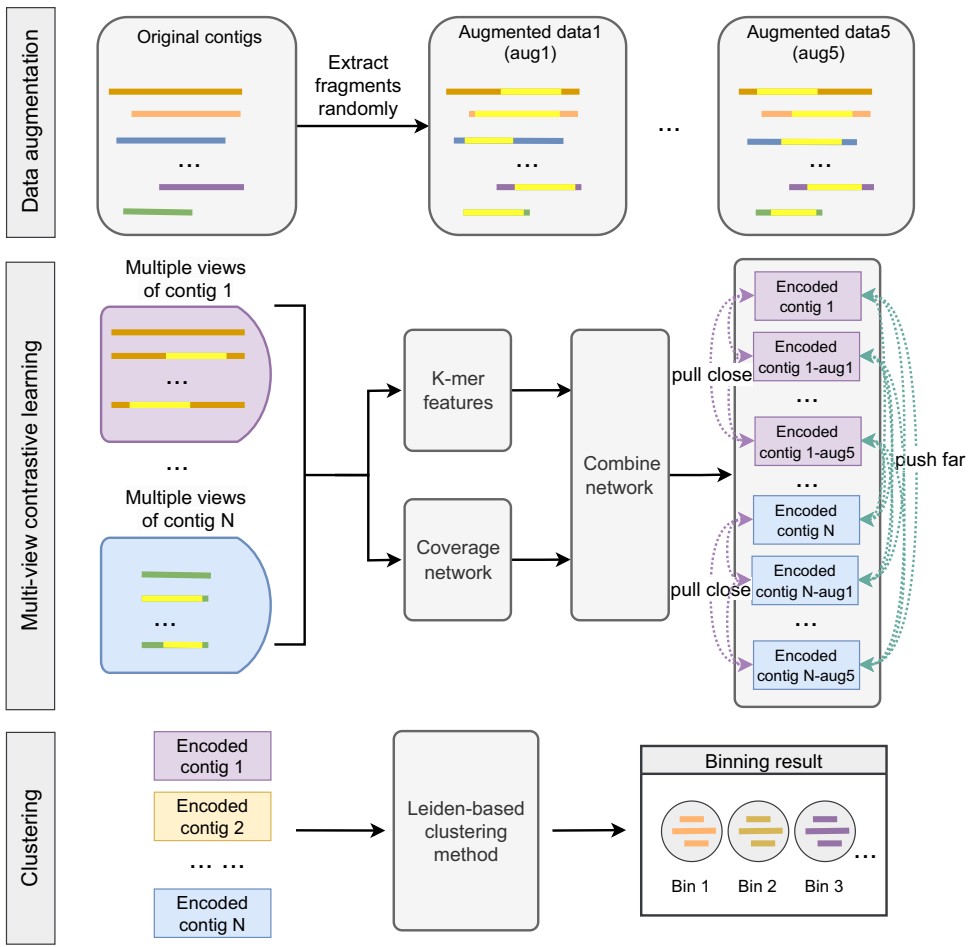

**Fig. 6 | The general workflow of COMEBin. The main steps of COMEBin are as follows (from top to bottom in the figure).** 1) Data augmentation: generate six views for each original contig, including one original and five augmented views. The yellow highlighted segments in the sequences indicate the extracted sequences during data augmentation. 2) Construct feature vectors and generate embeddings: obtain the features of the six views for each contig and generate low-dimensional representations using contrastive learning. 3) Clustering: generate binning results using the Leiden-based clustering method.

## Methods
### COMEBin
The framework of COMEBin is shown in Fig. 6, which is mainly divided into the following steps: 1) Data augmentation: construct five sets of augmented data by randomly extracting subsequences from the original contigs, resulting in six views for each original contig; 2) Construct feature vectors: construct nucleotide frequency and coverage features for each contig (including the original sequences and augmented sequences); 3) Contrastive learning: obtain low-dimensional representations suitable for binning with heterogeneous information based on multi-view contrastive learning, and 4) Clustering: generate binning results based on the Leiden community division algorithm[21].

Among them, the network structure used in contrastive learning includes two parts: 1) "Coverage network": process coverage features and 2) "Combine network": integrate the $k-$ mer features and the "Coverage network" to obtain representations containing heterogeneous information. More descriptions of COMEBin are as follows.

**Data augmentation.** Contiguous sequence fragments extracted from the same contig should belong to the same genome, ignoring potential errors caused by metagenomic sequence assembly. We proposed a data augmentation method based on the above assumptions. We randomly selected five continuous segments of not less than 1000bp for each contig to obtain five sets of augmented data. Consequently, we obtained six views for each contig, including one original view (original contigs) and five views corresponding to the augmented data.

**Construct feature vectors. Tetra-nucleotide frequencies (TNF)**
We utilized a sliding window of length $k$ with a stride of one to traverse the contig sequence, generating $k$-mer fragment sequences. The resulting $k$-mer frequency vector of dimension $T$ serves as a feature representation for the contig sequence, as described in ref. 35. In our binning algorithm, we set $k$ to 4 and $T$ to 136 (treating k-mers and their reverse complements as equivalent). The specific formulas employed are presented as follows.

$$F_i = (f_{i,1}, \ldots f_{i,j}, \ldots f_{i,136}),\tag{1}$$

where $f_{i,j}$ represents the frequency of the $j$-th $k$-mer feature of the $i$-th contig.

Based on Eq. (1), to avoid elements with zero values in the vector, we let $f'_{i,j} = f_{i,j} + 1$, and normalized $f'_{i,j}$ to remove the effect of different contig lengths. Let

$$q_{i,j} = \frac{f'_{i,j}}{\sum_{k=1}^{136} f'_{i,k}},\tag{2}$$

and

$$x_i^{(com)} = [q_{i,1}, \ldots, q_{i,136}],\tag{3}$$

where $x_i^{(com)} \in X^{(com)}$ represents the nucleotide frequency feature of the $i$-th contig.

#### Contig abundance (coverage)

Each contig assembled from reads of $M$ sequencing samples can be represented by a coverage vector ($2M$ dimensions) as follows. Let

$$C_i^{(mean)} = \left(c_{i,1}^{(mean)}, \ldots, c_{i,m}^{(mean)}, \ldots, c_{i,M}^{(mean)}\right), \tag{4}$$

where $c_{i,m}^{(mean)}$ represents the average of the number of reads covering each base of the $i$-th contig in the $m$-th sequencing sample. Define

$$C_i^{(std)} = \left(c_{i,1}^{(std)}, \ldots, c_{i,m}^{(std)}, \ldots, c_{i,M}^{(std)}\right), \tag{5}$$

where $c_{i,m}^{(std)}$ represents the standard deviation of the number of reads covering each base of the $i$-th contig in the $m$-th sequencing sample. We concatenated $C_i^{(mean)}$ and $C_i^{(std)}$ to form the coverage vector.

$$C_i = \left(C_i^{(mean)}, C_i^{(std)}\right). \tag{6}$$

To avoid zero vectors, we added a small fraction to each value of $c_{i,m}$, e.g., $c_{i,m}' = c_{i,m} + 1e-5$. The coverage vector is normalized across samples to remove the effect of different read counts from different sequencing samples.

$$x_{i,j}^{(cov)} = \frac{c_{i,m}'}{\max_{k=1}^{N} c_{k,m}'}, \tag{7}$$

where $N$ represents the number of contigs. Define

$$x_i^{(cov)} = \left[x_{i,1}^{(cov)}, \ldots, x_{i,m}^{(cov)}, \ldots, x_{i,2M}^{(cov)}\right], \tag{8}$$

where $x_i^{(cov)} \in X^{(cov)}$ represents the coverage feature of the $i$-th contig.

**Contrastive learning.** Contrastive learning obtains the representations of instances through unsupervised proxy tasks and optimizing contrastive losses[36]. By optimizing contrastive losses, the representations of similar instances are pulled closer, while those of dissimilar instances are pushed farther. Whether the instances are considered similar is based on the specific unsupervised proxy task. In our binning task, fragments extracted from the same original contig in the data augmentation step are regarded as similar instances.

The network structure used for contrastive learning is divided into two main modules: 1) a network used to obtain embedded representations of coverage features ("Coverage network") and 2) a network that integrates the two kinds of features ("Combine network"), which learns through contrastive learning to obtain embedded representations containing heterogeneous information.

**Coverage network ($f_{cov}$):** The "Coverage network" consists of a three-layer feed-forward neural network. The input layer comprises contig coverage features obtained through the method described in "Construct feature vectors". See Supplementary Table S3 for the hyper-parameters of the "Coverage network".

**Combine network ($f_{combine}$):** The "Combine network" consists of a three-layer feed-forward neural network. COMEBin normalizes the outputs of the "Coverage network" and concatenates them with $k$-mer features together as the input for the "Combine network". The output of the "Combine network" is an embedded representation containing heterogeneous information. The hyper-parameters of the network are shown in Supplementary Table S3.

**Objective function:** The COMEBin neural network uses the normalized temperature scale cross-entropy (NT-Xent) loss function[37,38] as the objective function. More details on the contrastive learning training process of COMEBin are given in Supplementary

Algorithm S1. Define

$$\ell(z_{i,k}, z_{i,k'}) = -\log \frac{\exp(\cos(z_{i,k}, z_{i,k'})/\tau)}{\sum_{s=1}^{N_{bs}} \mathbb{1}_{[s \neq i]} \exp(\cos(z_{i,k}, z_{s,k})/\tau) + \exp(\cos(z_{i,k}, z_{s,k'})/\tau)]} \tag{9}$$

where $z_{i,k}$ and $z_{i,k'}$ denotes the representation of the $k$-th and $k'$-th view of the $i$-th original contig, and $N_{bs}$ represents the batch size. The indicator function $\mathbb{1}_{[s \neq i]} \in \{0,1\}$ is defined to be one if and only if $s \neq i$. And $\cos(a,b)$ is defined as follows. Let

$$\cos(a,b) = \frac{a^T \cdot b}{|a| \cdot |b|}. \tag{10}$$

When under two views, the loss for each batch is as follows. Let

$$L_{2view} = \frac{1}{2N_{bs}} \sum_{i=1}^{N_{bs}} [\ell(z_{i,1}, z_{i,2}) + \ell(z_{i,2}, z_{i,1})], \tag{11}$$

where $z_{i,1}$ denotes the representation of the first view of the $i$-th original contig and $z_{i,2}$ denotes the representation of the second view of the $i$-th original contig.

We extended the above loss function under two views to the case where the number of views is $V$ ($V = 6$). Each batch has $N_{bs}$ original contigs, and each contig has a total of $V$ views. In each batch, the $V$ views of each contig are mutually positive samples, and all views from other contigs are mutually negative samples. The loss function for each batch is defined as follows. Let

$$L = -\frac{1}{N_{bs}V(V-1)} \sum_{i=1}^{N_{bs}} \sum_{v=1}^{V} \sum_{v_1=1, v_1 \neq v}^{V}$$
$$\log \frac{\exp(\cos(z_{i,v}, z_{i,v_1})/\tau)}{\exp(\cos(z_{i,v}, z_{i,v_1})/\tau) + \sum_{j=1}^{N_{bs}} \sum_{v_2=1}^{V} \mathbb{1}_{[j \neq i]} \exp(\cos(z_{i,v}, z_{j,v_2})/\tau)}, \tag{12}$$

where $z_{i,v}$ denotes the representation of the $v$-th view of the $i$-th original contig, and $\tau$ is a hyper-parameter, which represents the temperature coefficient used to adjust the emphasis on similar negative samples.

**Clustering.** After obtaining the trained "Coverage network" ($f_{coverage}$) and "Combine network" ($f_{combine}$) through contrastive multi-view learning, representations for the original contigs are generated. These embeddings are suitable for various clustering methods (see Supplementary Note), and, in our case, we applied the Leiden algorithm[21], an advanced community detection algorithm.

**(1) Leiden-based clustering**

Leiden[21] does not require a predefined number of clusters, making it well-suited for metagenomes where the exact number of genomes is unknown. First, we obtained the k-nearest neighbor graph according to the representations of the contigs and calculated the L2 distance efficiently using hnswlib package[39]. To focus on edges with low distances, we kept 50%, 80%, or 100% of edges with smaller values for subsequent clustering. Finally, the distance matrix is converted into a similarity map (see Eq. (13)). Define

$$S_{ij} = \exp\left(\frac{-\left|x_i - x_j\right|^2}{\sigma}\right), \tag{13}$$

where $S_{ij}$ represents the similarity between the $i$-th contig and the $j$-th contig, $x_i$ and $x_j$ represent the embeddings obtained by $f_{combine}$ of the two contigs, respectively, and $\sigma$ is a hyper-parameter.

Afterward, we utilized the community division algorithm Leiden[21] to divide the similarity map and obtain the clustering results. Notably,

our settings differ from the default configuration commonly used in Leiden in two aspects. Firstly, we assigned a unique cluster for each contig as the initial membership and designated contigs containing a particular single-copy gene (SCG) as fixed members, ensuring that the contigs with the designated SCG are clustered into separate clusters. This setting is motivated by the fact that single-copy genes exist as a single copy in a significant proportion (e.g., 97%) of the genomes within a specific phylum[22], and the contigs that harbor the same SCG are derived from distinct genomes. Secondly, we set the node size of each contig using its length, regarding each node as an aggregation of base pairs of the contig.

**(2) Choose the best result automatically**

The results of Leiden are sensitive to the resolution parameter and the $\sigma$ in Eq. (13). Moreover, it is hard to set the same parameters for different types of metagenomes. To address this, we ran Leiden using different parameters in parallel and selected the best result automatically. Parameters include $\sigma$ in Eq. (13) (0.05, 0.1, 0.15, 0.2, and 0.3), resolution parameters (1, 5, 10, 30, 50, 70, 90, and 110), and edge ratios (proportions of edges kept for clustering) of 50%, 80%, and 100%. We estimated the quality of the Leiden results with different parameters by estimating completeness and contamination of the bins using the similar method used in MetaBinner[40] (see Supplementary Note for more details). Through this approach, we obtained the estimated values across six metrics for each Leiden result, including the number of genomic bins with contamination levels below 5% or 10% and completeness levels exceeding 50%, 70%, or 90%. Finally, the binning result with the maximum sum of the estimated values across the six metrics was automatically chosen as the final output result. Following SemiBin1[19] and VAMB[15], we removed bins smaller than 200 kbp to get final bins.

## Binning modes

Co-assembly and single-sample binning are commonly utilized to benchmark the performance of binning methods[5,10,12,23]. Recently, VAMB[15] and SemiBin1[19] introduced the multi-sample binning, also referred to as "multisplit". In co-assembly binning, reads from all samples are pooled and assembled to generate the co-assembled contigs. Binning is performed using these co-assembled contigs and the coverage information obtained across all corresponding samples. In single-sample binning, reads from each sequencing sample are assembled separately, resulting in sample-specific contigs. Then, the sample-specific contigs and their respective coverage information are used for binning. Multi-sample binning is performed using the sample-specific contigs, but coverage information is derived by aligning the reads from all corresponding samples against the contigs. There are two available ways for multi-sample binning. VAMB[15] and CLMB[17] concatenate sample-specific contigs from all samples for binning and subsequently divide the bins based on the sample ID of each contig. SemiBin1[19] uses the sample-specific contigs for binning, and abundance information is aggregated across samples. In this paper, when referring to multi-sample binning, the mode of VAMB is employed for both VAMB and CLMB, while the mode of SemiBin1 is utilized for all other binning methods.

In this paper, unless otherwise specified, the binning mode used is the co-assembly.

## Benchmark datasets

The benchmark datasets were partitioned into training and holdout datasets following the experimental configuration outlined in ref. 15. The hyper-parameter selection for COMEBin was performed exclusively using the training datasets. We compared COMEBin with other state-of-the-art binning algorithms on sixteen datasets, including ten simulated and six real datasets. Sequence length distribution for the datasets is given in Supplementary Figs. S11 and S12. For more detailed dataset statistics, refer to Supplementary Tables S4 and S5.

**Simulated datasets.** The four simulated datasets used for training were provided by the organizers of the CAMI II challenge (https://data.cami-challenge.org): one from the mouse gut (CAMI Mouse gut) and three from human microbiomes (CAMI Skin, CAMI Airways, and CAMI Gastrointestinal tract (CAMI Gt) datasets). The gold standard cross-sample assemblies were used for binning. We used six benchmark datasets from CAMI II challenge[5] as the hold-out simulated datasets. These datasets include Marine GSA, Marine MA, Plant-associated GSA, Plant-associated MA, Strain-madness GSA, and Strain-madness MA. In this context, "GSA" refers to gold standard assembly, while "MA" refers to MEGAHIT assembly. As most of the compared binning methods can only handle contigs longer than 1000 bp, we kept contigs longer than 1000 base pairs for binning. The organizers of the CAMI II challenges provided raw simulated sequencing reads and ground truth annotations for contigs. More details are given in Supplementary Table S4.

**Real datasets.** We also used six real datasets for co-assembly binning comparison: the STEC dataset, the Water Group1 dataset, the Water Group2 dataset, the Water Group3 dataset, the MetaHIT dataset, and the MetaHIT (10-sample) dataset. The STEC dataset[41] contains sequencing data from 53 stool samples from the https://www.ebi.ac.uk/ena/browser/view/PRJEB1775 project of the European Nucleotide Sequence Archive. The three river datasets, namely Water Group1, Water Group2, and Water Group3, consist of 8, 5, and 7 sequencing samples, respectively. These datasets were employed to evaluate the performance of different binning methods on datasets with relatively small numbers of sequencing samples. They were sourced from the three groups of river sequencing samples of the https://www.ebi.ac.uk/ena/browser/view/PRJNA542960 project[29]. As for the MetaHIT dataset, a widely adopted large-scale multi-sample benchmark dataset, we utilized the advanced metagenomic assembly algorithm, MEGAHIT[42], to perform co-assembly on 264 sequencing samples from the MetaHIT consortium (Project: https://www.ebi.ac.uk/ena/browser/view/PRJEB2054)[43], generating contigs. To further demonstrate the usability of COMEBin on medium-scale datasets, we randomly selected ten samples from the MetaHIT sequencing samples, referred to as MetaHIT (10-sample).

The STEC dataset and Water Group1 dataset were used as the training datasets, while the remaining four real datasets served as holdout datasets. All datasets were co-assembled by the metagenomic sequence assembly tool MEGAHIT[42]. We kept contigs longer than 1000 base pairs for binning. Further details can be found in Supplementary Tables S5 and S6.

To further demonstrate the usability of COMEBin across different binning modes, we employed both single- and multi-sample binning to three real datasets: Water Group2, Water Group3, and MetaHIT (10-sample). To generate sample-specific contigs, the reads from each sequencing sample were individually assembled by MEGAHIT[42]. We kept contigs longer than 1000 base pairs for binning.

## Evaluation metrics

For the simulated datasets, we used AMBER[24] to calculate the number of high-quality genomes, ARI (bp), percentage of binned bp, accuracy (bp), and the harmonic mean of average purity (bp) and average completeness (bp): F1-score (bp).

We used CheckM2[25] to evaluate the completeness and contamination scores of the putative bins for the real datasets.

Similar to CheckM1[22] and CAMI II challenges[5], we refer bins with >50% completeness and <10% contamination as "moderate or higher" quality bins and bins with >90% completeness and < 5% contamination as near-complete genomes.

## Compared methods and experimental settings

We conducted a comprehensive comparison between COMEBin and eight state-of-the-art binning algorithms, namely CONCOCT 1.0.0[9],

MaxBin2 2.2.6[11], MetaBAT2 2.12.1[13], VAMB 4.1.3[15], CLMB[17], MetaDecoder 1.0.11[14], SemiBin1 1.0.0[19], and SemiBin2 1.5.1[20]. We ran CONCOCT, MaxBin2, and MetaBAT2 using the binning module of MetaWRAP 1.2.1[44]. The detailed commands for executing the compared binning methods are given at https://github.com/ziyewang/COMEBin_benchmark.

The reads were aligned to the contigs using BWA 0.7.17[45], and the per-position coverage information for the contigs, which was used in COMEBin, was calculated using BEDTools 2.30.0. Binning results on simulated and real datasets are evaluated by AMBER 2.0.3[24] and CheckM2 1.0.1[25]. To assess the taxonomic diversity captured by the different binning methods, we employed GTDB-Tk 2.3.0[26,27] with GTDB (release_214) to annotate the bins with >50% completeness and <5% contamination, using the classify_wf workflow with "–skip_ani_screen" option.

In our proposed method, we set the value of $\tau$ in Eq. (12) to 0.07 for assemblies with an N50 larger than 10,000, while for other assemblies, the value of $\tau$ was set to 0.15. The contrastive learning network was trained for 200 epochs using a batch size of 1024, and we implemented an early stopping mechanism. The hyper-parameters for each network module can be found in Supplementary Table S3. These hyper-parameters were determined based on the binning performance of COMEBin on the six training datasets.

COMEBin was developed with Python. The neural network was implemented using PyTorch. In the development of the Leiden-based clustering method, Python packages leidenalg (version 0.8.10), igraph (version 0.9.9), scikit-learn (version 0.22.1), and hnswlib (version 0.6.2) were used.

### Identification of ARGs and VFGs

We utilized the Resistance Gene Identifier (RGI version 6.0.2)[46] with default parameters to predict ARGs from the bins based on the Comprehensive Antibiotic Resistance Database (CARD version 3.2.7).

For the identification of virulence factor genes (VFGs), we predicted open reading frames (ORFs) in contigs using Prodigal (version 2.6.3). Subsequently, these ORFs were aligned against the VFDB core dataset protein sequences, accessible at http://www.mgc.ac.cn/VFs, utilizing BLASTP (version 2.14.1)[47]. We classified an ORF as a potential VFG if it exhibited a minimum of 80% identity over more than 70% of the length of its top hit in the database, following the methodology outlined in[29].

### Identification of biosynthetic gene clusters (BGCs)

Potential BGCs were identified in the moderate or higher quality bins using a secondary metabolite genome mining tool, antiSMASH (version 6.1.1)[34] with the "–genefinding-tool prodigal" option.

### Reporting summary

Further information on research design is available in the Nature Portfolio Reporting Summary linked to this article.

## Data availability

All the datasets used in this study are publicly available. The simulated datasets, including CAMI mouse gut, CAMI Airways, CAMI Gastrointestinal tract, CAMI Skin, Marine GSA, Marine MA, Plant-associated GSA, Plant-associated MA, Strain-madness GSA, and Strain-madness MA, were created by CAMI II Challenge[5]. These datasets can be accessed from the CAMI portal at https://data.cami-challenge.org. All the simulated datasets are also downloadable from their respective DOIs (CAMI mouse gut: 10.4126/FRL01-006421672; CAMI Airways, CAMI Gastrointestinal tract, and CAMI Skin: 10.4126/FRL01-006425518; Marine, Plant-associated and Strain-madness: 10.4126/FRL01-006425521). The MEGAHIT assemblies (MA) used in the CAMI II datasets have been archived on Zenodo at https://doi.org/10.5281/zenodo.10437337. The sequence data (STEC, Water Group, and

MetaHIT datasets) used in the study are publicly available in the ENA with study accessions PRJEB1775, PRJNA542960, and PRJEB2054. The sequencing reads of the BATS samples are publicly available in the NCBI with accession number PRJNA385855, and the corresponding assemblies are publicly available in the ENA with accession number PRJEB45951. For long-read datasets, the sequencing reads are publicly available in the National Genomics Data Center (NGDC) under the study accession PRJCA007414 (Runs: CRR344871 and CRR344872), in the ENA under the run accession SRR10963010, and in the NCBI under the run accession ERR9769275.

## Code availability

The COMEBin software is freely available at https://github.com/ziyewang/COMEBin under the GNU General Public License version v3. The COMEBin code used in this work[48] is also archived on Zenodo under https://doi.org/10.5281/zenodo.10158246. The commands for executing the binning methods, the assembler, and the analysis can be found at https://github.com/ziyewang/COMEBin_benchmark/tree/master/benchmark. The source codes for reproducing the figures in the manuscript, as well as the intermediate results, are available at: https://github.com/ziyewang/COMEBin_benchmark/tree/master/visualization.

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

## Acknowledgements

This work was supported by the National Natural Science Foundation of China (grant 62272105, S.F.Z.), Shanghai Municipal Science and Technology Major Project (grant 2018SHZDZX01, S.F.Z.), ZJ Lab and Shanghai Center for Brain Science and Brain-Inspired Intelligence Technology (S.F.Z.), the 111 Project (grant B18015, R.H.Y.) and China Postdoctoral Science Foundation (grant 2023M730650, Z.Y.W.).

## Author contributions

S.Z. conceived and supervised the project. S.Z. and Z.W. designed the study and the methodological framework. Z.W. and R.Y. implemented the methods. Z.W. and H.H. performed the experiments. Z.W., H.H., and W.L. analyzed the data. Z.W. drafted the paper. F.S. and S.Z. modified the paper. All authors agree to the content of the final paper.

## Competing interests

All authors declare no competing interests.
