## [Peer Review File · Nature Communications]

REVIEWER COMMENTS

Reviewer #1 (Remarks to the Author):

In this manuscript, Wang and colleagues introduce COMEBin, a new binning tool to recover Metagenome-Assembled Genomes (MAGs). COMEBin implements an innovative binning strategy based on neural networks to extract meaningful features from metagenomic data and a well-chosen clustering algorithm. The authors provide extensive benchmarks showing that in all situations, COMEBin recover more high-quality MAGs than other binning tools.

I have been able to download, install and test the tool on a different dataset than those used by the authors. I can confirm that COMEBin performance is outstanding and outperforms available state-of-the-art binning tools. On ten randomly selected samples of the PRJEB33338 bioproject (chicken caecal metagenomes), COMEBin recovers more HQ MAGs (completeness $\geq 90\%$ and contamination $< 5\%$) than SemiBin2 (+18%), MetaDecoder (+26%) and MetaBAT2 (+92%).

I have no doubts that COMEBin will be widely used by bioinformaticians working on metagenomic data and that the paper will be highly cited.

Yet, I have some concerns and suggestions that could be addressed in a revised version of the manuscript, hopefully.

Major

1) Overall, the results section looks a bit messy. Subsections order is not logic and those presenting "biology" related results (2.4, 2.5) should come last. Some subsections are very short (e.g 2.7, 2.8) and could be merged with others so that the manuscript structure will be more balanced.

2) Performance of state-of-the-art binning tools (MetaDecoder, SemiBin) seems poor especially for human gut samples. It surprises me a lot as it is contradictory with the benchmark I have performed for this review, all the other projects I have been working on and results presented by others (<https://www.nature.com/articles/s41467-022-29843-y/figures/3>, <https://microbiomejournal.biomedcentral.com/articles/10.1186/s40168-022-01237-8/figures/6>). In particular, MetaDecoder and SemiBin provide better results than MetaBAT2 for the vast majority of samples.

Consequently, the comparison with only MetaBAT2 in sections 2.4 and 2.5 those not seems fair to me. Please discuss this point.

3) Results – subsection 2.2: It seems that the authors used coverage information from multiple samples, not just one but it is not written explicitly. Please provide results for single coverage and multi coverage mode in two separate paragraphs and figures.

4) I am quite puzzled by the running times given in the manuscript. In my benchmarks, COMEBin was by far the slowest tool (> 10 hours per sample for COMEBin but 5 hours for SemiBin). The machine I used didn't have a GPU and a less powerful CPU, which may partly explain these differences.

Anyway, please also provide:

a) running times in CPU-only mode.

b) comparison with other tools especially those relying on neural networks (semibin, vamb)

Minor

1) The figures comparing the number of bins (Fig 1.A, Fig 2.A, Fig 3.A) among tools are made up of too many subplots making them difficult to read.

a) Please discard bins with completeness < 60% or contamination > 5% as checkm does not provide accurate estimates for low quality genomes (doi: 10.1101/2022.07.11.499243)

b) Use stacked bar plots (<https://academic.oup.com/view-large/figure/409579117/btad209f2.tif>) to combine on the same plots bins with different completeness.

2) Could the authors test a variant of ComeBin that uses different clustering methods (e.g. InfoMap implemented in SemiBin) to show the superiority of the Leiden algorithm?

3) Long-read sequencing is increasingly used as prices drop and read accuracy increase. Please provide a benchmark with such data. I am curious of the behavior of the data augmentation technique with a highly contiguous assembly.

4) Could the authors include SemiBin2 in their benchmark as it may provide better results than SemiBin1 ?

5) Could the authors switch from checkm1 to checkm2 to assess MAGs quality? Checkm2 was shown to provide more accurate estimates, especially for reduced-size genomes.

6) Please replace the figures presenting sum of ranks per metric of binners (Fig 1.B, Fig 2.B) by radar plots, which are more appropriate to compare metrics.

7) COMEBin relies on prokaryotic single-copy marker genes. Does it mean that it cannot recover eukaryotic genomes? Please address this point

What are the marker genes (those of checkm, gtdbtk?) How are they identified in the contigs?

8) The data augmentation technique looks somehow similar to the creation of must-link constraints in SemiBin and Coconet. Please discuss this point

9) Contrastive learning: It seems logic to pull close views from the same contig. Yet, pushing far views from different contigs is not intuitive as these contigs may belong to the same genome.

My guess is that it works because among all pairs of contigs, only a small fraction belong to the same genome in complex ecosystems.

Yet, I am afraid this strategy does not work with metagenomic samples from low complexity environments (= low species richness). Please discuss this point.

10) ""The Coverage network""consists of a three layer feed-forward neural network. The input layer is the contig coverage features obtained using the method of section 4.4.2.""

Here the input layer has $2 * M$ dimensions where M is the number of samples. The output layer has 128 dimensions according to table S8. In most cases, we measure contig coverage in a limited number of samples so $128 \gg 2 * M$. Does it make sense to you to have such a large output layer?

11) The installation is easy using conda but submission of the xml recipe on the bioconda channel would be welcomed.

12) Replace all occurrences of MetaBAT and MaxBin by MetaBAT2 and MaxBin2 resp.

13) Introduction: “vital in analyzing metagenomic” The adjective “vital” is excessive. Replace it with a more appropriate term.

14) Introduction: All the binning tools tested in this paper are presented, except MetaDecoder. Please add it.

15) “Conversely, COMEBin demonstrates insensitivity to the number of sequencing samples” This statement is ambiguous, as one can understand that calculating coverage information from more samples does not improve results. Please clarify.

16) Table S8 : input dimension of the combine network is said to be 256, however it seems to me that it should be $128+136 = 264$.

17) “Contiguous sequence fragments extracted from each contig should belong to the same genome as this contig, ignoring potential errors caused by metagenomic sequence assembly.” This sentence seems truncated

18) Add information about the programming language (Python) and libraries used (torch, sklearn, etc.)

Reviewer #2 (Remarks to the Author):

Major comments

Wang et al., present a novel method (COMEBin) for binning contigs generated by metagenome assembly. The method incorporates existing approaches (e.g., tetra-nucleotide frequencies, multi-

sample contig coverage, and contrastive learning) and utilizes novel approaches, especially for data augmentation and contig clustering. The authors provide a good deal of evaluations comparing their method versus many state-of-the-art approaches, and the evaluations are conducted on a number of simulated and real metagenomes. Based on these benchmarks, authors' approach appears quite promising. Still, I have a number of major and minor comments that I believe the authors should address in order to robustly and clearly show that their method is a substantial advance versus the state-of-the-art.

First, some critical evaluations are missing from the manuscript. For instance, the data augmentation approach would likely not be beneficial if most contigs are near the cutoff for minimum contig length of 1000 bp (both the original contig and the augmented contigs). The authors should evaluate the effectiveness of data augmentation as a function of N50. Second, the authors state that SemiBin cannot be used to bin contigs as done by VAMB or CLMB, but I believe that SemiBin can be used in the same manner as these other methods. If so, the authors should use the same "combined contig" binning approach for all methods in order to produce a fair evaluation. Lastly, in the Minor Comments, I list some minor evaluations that should be conducted.

Second, the authors evaluate BGC assembly but do not include BGC-targeting assembly methods. At the very least, the authors should include a comparison of using MEGAHIT + COMEBin versus biosyntheticSPAdes.

Third, the authors have left out critical information from the Methods, such as the details on how ARGs, VFGs, and BGCs were identified (see minor comments).

Fourth, the authors should include more thorough compute resource benchmarking. It is not clear how COMEBin compute resource requirements scale with critical parameters such as the number of features (contigs) or samples (metagenomes). Resource utilization should include CPU hours and memory, along with GPU usage & memory. The authors should directly compare resource utilization to other binning methods.

Lastly, the code associated with this work is currently lacking in clarity and robustness. The code cannot be easily installed via a standard software version manager (e.g., pip or conda), which creates barriers to integrating COMEBin into reproducible data analysis pipelines (e.g., Snakemake or Nextflow). COMEBin should at least be published on pypi, but I encourage the authors to also publish a Bioconda recipe. In regards to robustness, I encourage the authors to include unit/application testing and continuous integration (e.g., with GitHub Actions). In regards to clarity, the code is poorly documented (e.g., very little function-level documentation and no type hints). Finally, code for the

data analyses conducted in this work are not available (e.g., Jupyter notebooks showing how each figure and table in this manuscript were generated).

Minor comments

Abstract: Only mentioning MetaBat in the abstract gives the false impression that other state-of-the-art binning methods were not assessed.

Line 2: “;” => “by”

Line 7: What DNA fragments other than contigs would be binned?

Line 15: There are also taxonomy-based binning methods.

Line 23: The authors are actually referring to MaxBin2, correct?

Line 25: Rephrase as a scaling problem instead of “takes a long time”.

Line 26: The authors are actually referring to MetaBat2, correct?

Line 35: Briefly describe contrastive learning. The following sentences do not adequately define this methodology.

Line 64: Use parentheses to encompass the list of the methods.

Line 80: How is “pathogen” defined? Do the authors include opportunistic pathogens? Antibiotic resistance and BGCs in commensals are ignored?

Line 104: “ARI” => “Adjusted Rand Index (ARI)”

Line 104: What does “(bp)” mean?

Line 158: At what taxonomic level(s) (e.g., species or genus)?

Line 164: There is no need to start the section with “In this section”.

Line 168: These assembly methods are important for the reader to understand and should be described in the Introduction and not in the Supplemental Materials.

Line 197: I cannot find a description in the Methods section on how ARGs and VFGs were identified.

Line 219: Identification of BGCs (along with ARGs and VFGs) should be fully described in the Methods section.

Line 222: Are at least some of these “recovered BGCs” just artificially split across multiple contigs, which would thus inflate the perceived number of BGCs? In other words, how do we know that these BGCs are actually distinct versus just artificially split into partial BGCs?

Line 323: The authors should provide evidence or at least speculate on why COMEBin performed particularly well on the CAMI Mouse dataset. What is unique about that particular dataset? Ideally, the authors could identify unique attributes about the CAMI Mouse dataset and then simulate datasets with or without each target attribute in order to identify the dataset attribute(s) that result in high COMEBin performance.

Line 405: Which version (release) of the GTDB was used with GTDB-Tk?

Line 408: This sentence illustrates that the subsection on COMEBin should precede this subsection of the Methods, and should likely come first in the Methods section.

Line 409: What is the justification for changing tau, depending on the N50?

Line 412: What are the early stopping criteria?

Line 424: “on community division algorithm Leiden” => “on the Leiden community division algorithm”

Line 441: It could be helpful to explicitly state here that you used tetra-nucleotide frequencies (TNF).

Line 449: Why would zero vectors occur? The vector represents all possible 4-mers, while treating reverse compliments as equivalent.

Line 454: How were reads mapped to the contigs? What filtering of the raw read mapping was conducted (e.g., removing improper read pairs)? What software (and parameters) were used to obtain per-position (bp) coverage information?

Line 454: Is COMEBin suitable for single-end data? I’m assuming that it is not, but this should be explicitly stated.

Line 519: “got” (and “get”) is too colloquial. Please re-word.

Line 522: Explicitly state how edges were filtered/retained.

Line 547: What parameters were assessed? Which optimization method was used (e.g., grid search, random parameter search, bayesian optimization, etc)?

Figures 1a, 2a, & 3a: Reduce the plots to just “comp>90% & cont<5” and “comp>50% & cont<5”. The other plots can go in the supplemental materials.

Figures 1b & 2b: Plot by accuracy measure instead of by dataset (e.g., color by dataset & facet by accuracy measure). Show the actual accuracy values instead of the sum-of-ranks.

Figure 5: “contig and” => “contig, and”. Which ordination method was used? There is over-plotting of points, which can hide some inaccurate clustering. It can help to reduce the alpha (transparency) of the point colors in order to better visualize overlap.

Figure S2: Please define “known” and “unknown” for the context of this figure.

Table S1: “(10- sample)” is formatted oddly. Explicitly define “best results” in the table caption.

Table S3: Briefly summarize the ablation experiment in the table caption. Explicitly define “best results”.

Table S4: Briefly describe “views”. Explicitly define “best results”.

Table S5: Is “minutes” in wall-clock time or CPU minutes? Given that resource profiling is dependent on other processing occurring on the machine(s) at the time of profiling, multiple replicates should be conducted, with the aggregate and variance among replicates reported.

Table S6: “Dataset” => “Datasets”. Include the raw read lengths, read quality, sequencing platform, and library preparation method (if available).

Algorithm S1: “(com” => “(com)”. “(cov” => “(cov)”

Section S2.1: SemiBin can be used to bin contigs as done by VAMB and CLMB. See <https://github.com/BigDataBiology/SemiBin#easy-multi-samples-binning-mode>. It would be best to re-evaluate SemiBin using this “combined-contigs” method, as done with VAMB and CLMB.

Response Letter

Dear Editors and Reviewers,

We greatly appreciate your time and efforts in reviewing the original manuscript. We thank the reviewers for the highly pertinent and helpful comments. In the revision, we have conducted additional experiments and modified our manuscript accordingly by carefully considering the reviewers' suggestions and comments.

Here, we summarize the major changes made in the revised version of our manuscript.

1. Considering the reviewers' comments, we have included detailed descriptions of the three binning modes (co-assembly, single-sample, and multi-sample) in the "Introduction" section. Additionally, a subsection titled "Binning Modes" has been introduced within the "Methods" section. Furthermore, we have provided the binning results for three datasets in single- and multi-sample binning and the binning results for sixteen datasets in co-assembly binning.
2. Following the first reviewer's suggestion, we have included SemiBin2 in comparison.
3. Following the first reviewer's suggestion, we have replaced CheckM1 with CheckM2 to evaluate the performance of binning methods on real datasets.
4. We have included more binning methods for comparison in the analysis to identify potential pathogenic antibiotic-resistant bacteria (PARB) or metagenome-assembled genomes that contain potential BGCs.
5. We have improved the user-friendliness of our GitHub repository and published a Bioconda recipe, taking into account the reviewers' comments.

In the following, we present one-by-one responses to the reviewers' comments.

Reviewer #1:

Comment — In this manuscript, Wang and colleagues introduce COMEBin, a new binning tool to recover Metagenome-Assembled Genomes (MAGs). COMEBin implements an innovative binning strategy based on neural networks to extract meaningful features from metagenomic data and a well-chosen clustering algorithm. The authors provide extensive benchmarks showing that in all situations, COMEBin recover more high-quality MAGs than other binning tools.

I have been able to download, install and test the tool on a different dataset than those used by the authors. I can confirm that COMEBin performance is outstanding and outperforms available state-of-the-art binning tools. On ten randomly selected samples of the PRJEB33338 bioproject (chicken caecal metagenomes), COMEBin recovers more HQ MAGs (completeness $\geq 90\%$ and contamination $< 5\%$) than SemiBin2 (+18%), MetaDecoder (+26%) and MetaBAT2 (+92%).

I have no doubts that COMEBin will be widely used by bioinformaticians working on metagenomic data and that the paper will be highly cited.

Yet, I have some concerns and suggestions that could be addressed in a revised version of the manuscript, hopefully.

Response: Thank you for your positive feedback and helpful comments. We sincerely appreciate your effort in testing COMEBin.

Major comments

Comment — 1) Overall, the results section looks a bit messy. Subsections order is not logic and those presenting “biology” related results (2.4, 2.5) should come last. Some subsections are very short (e.g 2.7, 2.8) and could be merged with others so that the manuscript structure will be more balanced.

Response: We have relocated the “biology-related results” initially found in sections 2.4 and 2.5 to the end of the “Results” section. Furthermore, we have merged the original Section 2.7 (“The effect of the amount of augmented data on the results”) with Section 2.6 (“Contrastive multi-view learning improves binning results”).

Comment — 2) Performance of state-of-the-art binning tools (MetaDecoder, SemiBin) seems poor especially for human gut samples. It surprises me a lot as it is contradictory with the benchmark I have performed for this review, all the other projects I have been working on and results presented by others (<https://www.nature.com/articles/s41467-022-29843-y/figures/3>, <https://microbiomejournal.biomedcentral.com/articles/10.1186/s40168-022-01237-8/figures/6>). In particular, MetaDecoder and SemiBin provide better results than MetaBAT2 for the vast majority of samples. Consequently, the comparison with only MetaBAT2 in sections 2.4 and 2.5 those not seems fair to me. Please discuss this point.

Response: The binning performance can be influenced by various factors, including dataset characteristics such as N50, the number of sequencing samples, and the number of contigs, as well as the binning modes for the benchmark. Different datasets and binning modes may result in different performance. In our manuscript, the results presented in the sections titled “COMEBin outperforms other binning methods on the simulated datasets” and “COMEBin outperforms other binning methods on the real datasets” were evaluated specifically under the co-assembly binning. And the co-assembly binning mode is commonly utilized to benchmark the performance of binning methods, including CAMI I and II [1–4]. And

the binning performance of SemiBin1 [5] on real datasets and MetaDecoder [6] were mainly evaluated in single- and multi-sample binning in their papers [5, 6].

This revised version describes the three binning modes in the “Introduction” section and the “Binning modes” subsection (see L73-L80, L540-L561). Furthermore, we have included evaluations on three datasets using the single-sample and multi-sample binning. Additionally, we have included MetaDecoder and SemiBin2 in the comparative analysis presented in the sections titled “COMEBin assists analysis of potential pathogenic antibiotic-resistant bacteria (PARB)” and “COMEBin helps to recover moderate or higher quality bins carrying potential BGCs,” originally found in sections 2.4 and 2.5.

Comment — 3) Results – subsection 2.2: It seems that the authors used coverage information from multiple samples, not just one but it is not written explicitly. Please provide results for single coverage and multi coverage mode in two separate paragraphs and figures.

Response: The results presented in subsection 2.2 were evaluated using the co-assembly binning mode. Furthermore, we have evaluated three datasets using single-sample and multi-sample binning modes, and the results are summarized in subsection 2.3 (see L209-L215) and Supplementary Fig. S6.

Comment — 4) I am quite puzzled by the running times given in the manuscript. In my benchmarks, COMEBin was by far the slowest tool (> 10 hours per sample for COMEBin but 5 hours for SemiBin). The machine I used didn’t have a GPU and a less powerful CPU, which may partly explain these differences. Anyway, please also provide:

- a) running times in CPU-only mode.
- b) comparison with other tools especially those relying on neural networks (semibin, vamb)

Response: We have included SemiBin2 and VAMB in the comparison, evaluating their performance in both CPU-only and GPU modes. Additionally, we added two datasets to the comparison, allowing users to assess the tool’s suitability for their specific use case. Moreover, addressing the second reviewer’s comments, we ran each tool three times on each dataset and computed the average running time and memory usage. The corresponding results are presented in the section “Running time and memory usage” (see L282-L300) and Supplementary Table S1.

Minor comments

Comment — 1) The figures comparing the number of bins (Fig 1.A, Fig 2.A, Fig 3.A) among tools are made up of too many subplots making them difficult to read.

- a) Please discard bins with completeness < 60% or contamination > 5% as checkm does not provide accurate estimates for low quality genomes (doi: 10.1101/2022.07.11.499243)
- b) Use stacked bar plots (<https://academic.oup.com/view-large/figure/409579117/btad209f2.tif>) to combine on the same plots bins with different completeness.

Response: We have represented the number of bins using stacked bar plots in Fig 1.a, Fig 2.a, and Fig 3.a. To our knowledge, the low-quality genomes denote the MAGs with “cont < 10%” and “comp < 50%” as defined in the paper referenced in this comment [7] (doi: 10.1101/2022.07.11.499243). Considering the comments from you and the other reviewer on Figures 1-3, we have kept bins with completeness > 50% and contamination < 5% and used stacked bar plots to present the bin numbers.

Comment — 2) Could the authors test a variant of ComeBin that uses different clustering methods (e.g. InfoMap implemented in SemiBin) to show the superiority of the Leiden algorithm?

Response: We have conducted experiments with different variants of COMEBin, replacing the Leiden-based clustering method with Infomap, as implemented in SemiBin1. Additionally, we employed k-means

and weighted k-means for clustering, utilizing the embeddings as features, and determined bin numbers based on single-copy genes. In “weighted k-means”, we assigned the weight for each contig based on its length. For Infomap, we used the same graphs converted from the embeddings as inputs, following the same methodology for automatically selecting the best result as originally used in COMEBin. Parameters to generate the graphs include σ in equation 13 with values of 0.05, 0.1, 0.15, 0.2, and 0.3, along with edge ratios (proportions of edges kept for clustering) with values of 50%, 80%, and 100%. COMEBin demonstrates better performance compared to its variants. The detailed results are shown in Supplementary Fig. S13. We have added the related descriptions in the Supplementary Note (“Comparison of variants of COMEBin using different clustering methods”).

Comment — 3) Long-read sequencing is increasingly used as prices drop and read accuracy increase. Please provide a benchmark with such data. I am curious of the behavior of the data augmentation technique with a highly contiguous assembly.

Response: In this revised version, we have compared COMEBin with SemiBin2, MetaDecoder, and MetaBAT2 using four long-read sequencing datasets. Notably, three of these datasets were also used in the SemiBin2 manuscript [8]. An important aspect highlighted in the SemiBin2 paper is the distinct clustering methods tailored for short reads and third-generation sequencing data. To ensure a fair comparison, we ran SemiBin2 in both its default and long-read modes (using the option “-sequencing-type=long_read”). According to the results shown in Supplementary Fig. S10, SemiBin2 (long-read mode) performs best, followed by COMEBin. Long-read sequencing typically yields highly contiguous assemblies, resulting in fewer contigs and smaller bins (measured by the number of contigs) [8]. As we did not design a binning algorithm specifically for long-read sequencing data, we discuss the related findings in the “Discussion” section (see L382-L391). In future research, we plan to design a binning algorithm specifically for long-read sequencing data and explore the application of the multi-view contrastive learning method on third-generation data.

Comment — 4) Could the authors include SemiBin2 in their benchmark as it may provide better results than SemiBin1?

Response: We have included SemiBin2 [8] in the benchmark.

Comment — 5) Could the authors switch from checkm1 to checkm2 to assess MAGs quality? Checkm2 was shown to provide more accurate estimates, especially for reduced-size genomes.

Response: We have switched from CheckM1 to CheckM2 [7] to assess MAGs quality.

Comment — 6) Please replace the figures presenting sum of ranks per metric of binners (Fig 1.B, Fig 2.B) by radar plots, which are more appropriate to compare metrics.

Response: We have replaced the figures presenting the sum of ranks per metric of binners (Fig 1.b, Fig 2.b) with radar plots as shown in Supplementary Fig. S1 and S2. We also show the actual accuracy values instead of the sum-of-ranks in Fig 1.b, Fig 2.b, following the comments of the second reviewer.

Comment — 7) COMEBin relies on prokaryotic single-copy marker genes. Does it mean that it cannot recover eukaryotic genomes? Please address this point
What are the marker genes (those of checkm, gtdbtk?) How are they identified in the contigs?

Response: COMEBin can recover eukaryotic genomes. Although we estimated the quality of the Leiden results with different parameters by estimating completeness and contamination based on prokaryotic SCGs, COMEBin can still recover eukaryotic genomes. Take the Plant-associated GSA dataset as an

example, which includes fungal genomes (the coverage information of the fungal genomes). In terms of bins with “comp > 50%” and “cont < 5%”, COMEBin can recover two fungal genomes, with the best-case method (MetaDecoder) recovering five fungal genomes and the worst-case method (MetaBAT2) recovering 0 fungal genomes. COMEBin can recover genomes originating from domains other than bacteria and archaea, although it does not show superiority in this case.

The marker genes are from domain-specific (bacterial and archaeal) marker sets of CheckM. We ran CheckM for one binning result and obtained the contigs having the single-copy genes. The relevant statements can be found in the Supplementary Note under the section titled “Estimating completeness and contamination of the bins.”

Comment — 8) The data augmentation technique looks somehow similar to the creation of must-link constraints in SemiBin and Coconet. Please discuss this point

Response: SemiBin constructs pairwise must-link constraints by splitting long contigs (eg. longer than 4000 bp) into two equal-length segments. CoCoNet [9] splits the contigs longer than 2048 bp into regularly spaced fragments of size 1024 bp (step: 128 bp), and fragment pairs from the same contigs are regarded as must-link constraints (named positive pairs in CoCoNet).

In contrast, we randomly selected five continuous segments of no less than 1000bp for each contig to obtain five augmented data sets. To accommodate different types of datasets, we did not set the fixed length for sequence fragments. In addition, our method supports modifying the number of segments extracted from each sequence, thereby altering the quantity of augmented data.

Comment — 9) Contrastive learning: It seems logic to pull close views from the same contig. Yet, pushing far views from different contigs is not intuitive as these contigs may belong to the same genome. My guess is that it works because among all pairs of contigs, only a small fraction belong to the same genome in complex ecosystems. Yet, I am afraid this strategy does not work with metagenomic samples from low complexity environments (= low species richness). Please discuss this point.

Response: In our opinion, contrastive learning works for two main reasons. First, the relative distance between sequences from the same genome is smaller than the distance between sequences from different genomes, making clustering effective. Second, in each training batch, only a small fraction of contigs belong to the same genome. To demonstrate that COMEBin works in low-complexity datasets, we randomly selected ten genomes from the CAMI skin and CAMI mouse gut datasets for testing. As shown in Supplementary Fig. S9, COMEBin recovers the most near-complete genomes on these two datasets.

Comment — 10) “The Coverage network” consists of a three layer feed-forward neural network. The input layer is the contig coverage features obtained using the method of section 4.4.2.” Here the input layer has $2 \cdot M$ dimensions where M is the number of samples. The output layer has 128 dimensions according to table S8. In most cases, we measure contig coverage in a limited number of samples so $128 \gg 2 \cdot M$. Does it make sense to you to have such a large output layer?

Response: We set a large output layer to make it compatible with different datasets and minimize the need for users to tune this super-parameter. We have adjusted the output layers of the “Coverage network” to 64, 32, and 16, resulting in COMEBin (dim: 64), COMEBin (dim: 32), and COMEBin (dim: 16). We then evaluated their performance on six datasets. The results are given in Figure R1. We computed the sum of rankings for different methods across the six datasets (CAMI Gt, CAMI Airways, CAMI skin, CAMI Mouse gut, STEC, and Water Group1 datasets) regarding the number of recovered bins with > 90% and > 50% completeness and < 5% contamination. The method with the highest value on each dataset gets a score of 0; the second best gets a score of 1, and so on. Additionally, we summed the proportions of each method’s decrease compared to the optimal result across the metrics. The results in Table R1

indicate that despite the marginal differences, the method with an output layer of 128 dimensions, namely COMEBin, achieves the best overall results. Therefore, we retain the original parameter settings.

Table R1 Comparison of COMEBin using different output dimensions for the “Coverage network”.

	Sum of ranks				Sum of decreased values compared to the best			
	#bins comp <5% cont)	(>90% >50% cont)	#bins comp <5% cont)	(>90% >50% cont)	#bins comp <5% cont)	(>90% >50% cont)	#bins comp <5% cont)	(>90% >50% cont)
COMEBin	6		5		-0.14		-0.03	
COMEBin (dim: 64)	7		11		-0.29		-0.18	
COMEBin (dim: 32)	13		11		-0.53		-0.19	
COMEBin (dim: 16)	6		7		-0.16		-0.10	

Comment — 11) The installation is easy using conda but submission of the xml recipe on the bioconda channel would be welcomed.

Response: We have published a bioconda recipe (<https://github.com/ziyewang/COMEBin>).

Comment — 12) Replace all occurrences of MetaBAT and MaxBin by MetaBAT2 and MaxBin2 resp.

Response: We have replaced all occurrences of MetaBAT and MaxBin by MetaBAT2 and MaxBin2, respectively.

Comment — 13) Introduction: “vital in analyzing metagenomic” The adjective “vital” is excessive. Replace it with a more appropriate term.

Response: We have replaced it with “important”.

Comment — 14) Introduction: All the binning tools tested in this paper are presented, except MetaDecoder. Please add it.

Response: We have added it into the “Introduction” section (see L30-L34) as follows:

“The recently developed MetaDecoder [6] is a two-layer contig binning model using a modified Dirichlet Gaussian mixture model to create preliminary clusters based on k-mer frequency and coverage. Then, it employs a semi-supervised k-mer frequency probabilistic model and a modified Gaussian mixture model for coverage to generate pure clusters.”

Comment — 15) “Conversely, COMEBin demonstrates insensitivity to the number of sequencing samples” This statement is ambiguous, as one can understand that calculating coverage information from more samples does not improve results. Please clarify.

Response: We have modified it into:

“Conversely, COMEBin still performs well on the datasets with few samples.”

Comment — 16) Table S8 : input dimension of the combine network is said to be 256, however it seems to me that it should be $128+136 = 264$.

Response: Thanks for this comment. We have corrected it.

Comment — 17) “Contiguous sequence fragments extracted from each contig should belong to the

same genome as this contig, ignoring potential errors caused by metagenomic sequence assembly.” This sentence seems truncated

Response: We have modified it into:

“Contiguous sequence fragments extracted from the same contig should belong to the same genome, ignoring potential errors caused by metagenomic sequence assembly.”

Comment — 18) Add information about the programming language (Python) and libraries used (torch, sklearn, etc.)

Response: We have added the related descriptions in the “Compared methods and experimental settings” section (see L648-651) as follows:

“COMEBin was developed with Python. The neural network was implemented using PyTorch. In the development of Leiden-base clustering method, Python packages leidenalg (version 0.8.10), igraph (version 0.9.9), scikit-learn (version 0.22.1), and hnswlib (version 0.6.2) were used.”

Reviewer #2:**Major comments**

Comment — Wang et al., present a novel method (COMEBin) for binning contigs generated by metagenome assembly. The method incorporates existing approaches (e.g., tetra-nucleotide frequencies, multi-sample contig coverage, and contrastive learning) and utilizes novel approaches, especially for data augmentation and contig clustering. The authors provide a good deal of evaluations comparing their method versus many state-of-the-art approaches, and the evaluations are conducted on a number of simulated and real metagenomes. Based on these benchmarks, authors' approach appears quite promising. Still, I have a number of major and minor comments that I believe the authors should address in order to robustly and clearly show that their method is a substantial advance versus the state-of-the-art.

Response: Thank you for your thoughtful review of our manuscript and helpful comments.

Comment — First, some critical evaluations are missing from the manuscript. For instance, the data augmentation approach would likely not be beneficial if most contigs are near the cutoff for minimum contig length of 1000 bp (both the original contig and the augmented contigs). The authors should evaluate the effectiveness of data augmentation as a function of N50. Second, the authors state that SemiBin cannot be used to bin contigs as done by VAMB or CLMB, but I believe that SemiBin can be used in the same manner as these other methods. If so, the authors should use the same “combined contig” binning approach for all methods in order to produce a fair evaluation. Lastly, in the Minor Comments, I list some minor evaluations that should be conducted.

Response:

Supplementary Fig. S11 and S12 show the sequence length distribution of simulated and real datasets. For the real datasets, the proportion of sequences shorter than 1500bp ranges from 0.49 to 0.61. The proportion of sequences shorter than 1500bp of CAMI Mouse gut reaches 0.76. COMEBin works well on these datasets. In our experience, it's rare for a dataset to consist almost entirely of sequences around 1000bp and proportion of sequences shorter than 1500bp in real datasets usually does not exceed 70%.

As for the second comment, our descriptions in the main text may have caused some misunderstanding. We didn't mean that SemiBin can not be used in multi-sample mode. There are two available ways for this mode. VAMB [10] and CLMB [11] concatenate sample-specific contigs from all the samples for binning and split the bins by sample information. SemiBin's multi-sample mode differs slightly from VAMB's. As stated in the paper of SemiBin, SemiBin [5] uses the sample-specific contigs for binning, and abundance information is aggregated across samples. This approach can be easily adopted by other binners without modifying the codes. Therefore, we adopted this approach and evaluated all binning methods, excluding VAMB and CLMB. Due to the indication by the VAMB authors on GitHub that VAMB is not quite suitable for small datasets (<https://github.com/RasmussenLab/vamb>), and CLMB is based on VAMB's work, we retained their original multi-sample mode method. In summary, when referring to multi-sample binning in the manuscript, the mode of VAMB is employed for both the VAMB and CLMB, while the mode of SemiBin1 is utilized for all other binning methods.

In our original manuscript, the results presented in the sections titled “COMEBin outperforms other binning methods on the simulated datasets” and “COMEBin outperforms other binning methods on the real datasets” were evaluated under the co-assembly binning. And the co-assembly binning is commonly utilized to benchmark the performance of binning methods, including CAMI I and II [1–4]. We have detailed descriptions of the three binning modes in the sections “Introduction” and “Binning modes”

(see L73-L80, L540-L561), and included evaluations on three datasets using the single-sample and multi-sample binning modes. Additionally, the commands to run the binning methods in each mode are given in https://github.com/ziyewang/COMEBin_benchmark/tree/master/benchmark.

Comment — Second, the authors evaluate BGC assembly but to not include BGC-targeting assembly methods. At the very least, the authors should include a comparison of using MEGAHIT + COMEBin versus biosyntheticSPAdes.

Response: In the manuscript, we mainly focus on exploring the repertoire of secondary metabolite biosynthetic gene clusters (BGCs) encoded within the metagenomic assembly genomes (MAGs) [12]. We utilized various binning methods on the same assembled contigs to generate MAGs, subsequently identifying BGCs based on these MAGs. We aim to demonstrate that the combination of assembled contigs and COMEBin produces more high-quality bins carrying BGCs than those with other binning methods. Because the BGC-targeting assembly methods, such as biosyntheticSPAdes, could not recover MAGs from metagenomic samples, we couldn't directly compare them with the pipeline using COMEBin in this study.

Comment — Third, the authors have left out critical information from the Methods, such as the details on how ARGs, VFGs, and BGCs were identified (see minor comments).

Response: We have added subsections “Identification of ARGs and VFGs” (see L652-L662) and “Identification of biosynthetic gene clusters (BGCs)” (see L663-L666) into the manuscripts. More details are given in the response to the related minor comments.

Comment — Fourth, the authors should include more thorough compute resource benchmarking. It is not clear how COMEBin compute resource requirements scale with critical parameters such as the number of features (contigs) or samples (metagenomes). Resource utilization should include CPU hours and memory, along with GPU usage and memory. The authors should directly compare resource utilization to other binning methods.

Response: We have included SemiBin2 and VAMB in the comparison, evaluating their performance in both CPU-only and GPU modes. Additionally, we added two datasets to the comparison, allowing users to assess the tool's suitability for their specific use case. Moreover, we ran each tool three times on each dataset and computed the average running time and memory usage. The corresponding results are presented in the section “Running time and memory usage” (see L282-L300) and Supplementary Table S1.

Comment — Lastly, the code associated with this work is currently lacking in clarity and robustness. The code cannot be easily installed via a standard software version manager (e.g., pip or conda), which creates barriers to integrating COMEBin into reproducible data analysis pipelines (e.g., Snakemake or Nextflow). COMEBin should at least be published on pypi, but I encourage the authors to also publish a Bioconda recipe. In regards to robustness, I encourage the authors to include unit/application testing and continuous integration (e.g., with GitHub Actions). In regards to clarity, the code is poorly documented (e.g., very little function-level documentation and no type hints). Finally, code for the data analyses conducted in this work are not available (e.g., Jupyter notebooks showing how each figure and table in this manuscript were generated).

Response: We have improved the user-friendliness of our GitHub repository (<https://github.com/ziyewang/COMEBin>) and published a Bioconda recipe. In addition, we have added function-level documentation and type hints for the codes. The commands for running the binning methods, the assembler, and the analyzing are given in

https://github.com/ziyewang/COMEBin_benchmark/tree/master/benchmark/README.md. The codes for reproducing figures in the manuscript and intermediate results are also given at https://github.com/ziyewang/COMEBin_benchmark/tree/master/visualization.

Minor comments

Comment — Abstract: Only mentioning MetaBat in the abstract gives the false impression that other state-of-the-art binning methods were not assessed.

Response: Thank you for your kind reminder. We have revised the relevant descriptions and refrained from mentioning MetaBAT in the abstract.

Comment — Line 2: “,” => “by”

Response: We have made the change accordingly.

Comment — Line 7: What DNA fragments other than contigs would be binned?

Response: Some early binning methods directly bin sequencing reads [13].

Comment — Line 15: There are also taxonomy-based binning methods.

Response: We mainly categorize the binning methods based on the sequence features they utilize; therefore, we did not discuss taxonomy-based binning methods.

Comment — Line 23: The authors are actually referring to MaxBin2, correct?

Response: We referred to MaxBin2. We have replaced all occurrences of MaxBin with MaxBin2.

Comment — Line 25: Rephrase as a scaling problem instead of “takes a long time”

Response: We have modified the descriptions as follows:

“Moreover, MaxBin2 can be computationally prohibitive on the datasets with a substantial number of sequencing samples.”

Comment — Line 26: The authors are actually referring to MetaBat2, correct?

Response: We referred to MetaBAT. We have included relevant descriptions of MetaBAT2 as follows:
“MetaBAT2 [14] combines two distance scores by calculating their geometric mean.”

Comment — Line 35: Briefly describe contrastive learning. The following sentences do not adequately define this methodology.

Response: We have added the descriptions as follows (see L41-L44):

“Contrastive learning is a self-supervised learning technique used for learning an informative representation of the input data by bringing similar instances closer together while pushing dissimilar instances farther apart [15].”

Comment — Line 64: Use parentheses to encompass the list of the methods.

Response: We have used parentheses to encompass the list of methods.

Comment — Line 80: How is “pathogen” defined? Do the authors include opportunistic pathogens? Antibiotic resistance and BGCs in commensals are ignored?

Response: We are referring to potential pathogenic antibiotic-resistant bacteria (PARB) that carry both antibiotic resistance genes (ARGs) and virulence factor genes (VFGs), as defined in a previous study [16]. We did not consider opportunistic pathogens. We focused on antibiotic resistance and virulence factor genes to identify potential pathogenic antibiotic-resistant bacteria (PARB). However, in analyzing potential biosynthetic gene clusters (BGCs), we did not consider antibiotic resistance and BGCs in commensals.

Comment — Line 104: “ARI” => “Adjusted Rand Index (ARI)”

Response: We have modified it accordingly.

Comment — Line 104: What does “(bp)” mean?

Response: The notation “(bp)” indicates that the evaluations are based on base pair counts as done in [3, 4, 17]. We have added the statements in the manuscript (see L127-L128).

Comment — Line 158: At what taxonomic level(s) (e.g., species or genus)?

Response: The x-axis of the figures represents taxa ranging from the species level to the domain level.

Comment — Line 164: There is no need to start the section with “In this section”.

Response: We have modified it accordingly.

Comment — Line 168: These assembly methods are important for the reader to understand and should be described in the Introduction and not in the Supplemental Materials.

Response: We have included descriptions of the three binning modes (co-assembly, single-sample, and multi-sample) in the “Introduction” section (see L73-L80). Additionally, we have added a subsection titled “Binning modes” within the “Methods” section (see L540-L561).

Comment — Line 197: I cannot find a description in the Methods section on how ARGs and VFGs were identified.

Response: In the initial manuscript, we utilized PathoFact [18], a pipeline designed for predicting virulence factor genes (VFGs) and antimicrobial resistance genes (ARGs) in metagenomic data, to identify ARGs and VFGs within the bins. However, during the process of preparing the revised manuscript, we encountered an unresolved issue (<https://gitlab.lcsb.uni.lu/laura.denies/PathoFact/-/issues/94>) when executing PathoFact on the candidate bins selected based on the CheckM2 evaluations. We have changed the method of identifying ARGs and VFGs in the revised version. In this updated version, we used the Resistance Gene Identifier (RGI version 6.0.2) [19] with default parameters to predict ARGs from nucleotide data based on the Comprehensive Antibiotic Resistance Database (CARD version 3.2.7).

For the identification of virulence factor genes (VFGs), we predicted open reading frames (ORFs) in contigs using Prodigal (version 2.6.3). Subsequently, these ORFs were aligned against the VFDB

core dataset protein sequences, accessible at <http://www.mgc.ac.cn/VFs>, utilizing BLASTP (version 2.14.1) [20]. An ORF was considered a potential VFG if it exhibited a minimum of 80% identity over more than 70% of the length of its top hit in the database, following the methodology outlined in [16].

We have added the related descriptions in the “Identification of ARGs and VFGs” subsection (see L652-L662).

Comment — Line 219: Identification of BGCs (along with ARGs and VFGs) should be fully described in the Methods section.

Response: Potential BGCs were identified in the moderate or higher bins using a secondary metabolite genome mining tool, antiSMASH (version 6.1.1) [21] with the “-genefinding-tool prodigal” option. We have added the related descriptions in the “Identification of biosynthetic gene clusters (BGCs)” subsection (see L663-L666).

Comment — Line 222: Are at least some of these “recovered BGCs” just artificially split across multiple contigs, which would thus inflate the perceived number of BGCs? In other words, how do we know that these BGCs are actually distinct versus just artificially split into partial BGCs?

Response: Thanks for raising this point. We have checked the outputs of antiSMASH and found that most of the identified BGCs were located at the edge of the contigs. In this revision, we have added the results of the number of moderate or higher bins carrying at least one BGC to reduce the impact of the overestimation of BGC quantity. The relevant results are shown in Figure 5e.

Comment — Line 323: The authors should provide evidence or at least speculate on why COMEBin performed particularly well on the CAMI Mouse dataset. What is unique about that particular dataset? Ideally, the authors could identify unique attributes about the CAMI Mouse dataset and then simulate datasets with or without each target attribute in order to identify the dataset attribute(s) that result in high COMEBin performance.

Response: The original statement was not precise; we have removed the relevant content. In addition to the improvement of 24.3% in near-complete genomes recovery achieved by COMEBin for the CAMI Mouse gut dataset compared to the second-best method, it also exhibits substantial enhancements of 29.9%, 18.2%, 54.2%, 25.9%, 57.7%, and 54.1% for the CAMI skin, Marine GSA, STEC, Water group2, MetaHIT (10-sample; single-sample mode), and Water Group3 (multi-sample binning) datasets, respectively. These improvements encompass diverse datasets, including those with a sample size of over 50, single-sample, simulated, and real datasets.

In this paper, we aim to develop a binning algorithm capable of effectively handling various types of metagenomic data, and the results above and in the manuscript also illustrate this point.

Comment — Line 405: Which version (release) of the GTDB was used with GTDB-Tk?

Response: We used GTDB (release_214) and added the descriptions to the manuscript.

Comment — Line 408: This sentence illustrates that the subsection on COMEBin should precede this subsection of the Methods, and should likely come first in the Methods section.

Response: We have relocated the subsection on COMEBin to the beginning of the “Methods” section.

Comment — Line 409: What is the justification for changing tau, depending on the N50?

Response: We determined the candidate values of τ based on COMEBin’s performance on the training

datasets. Based on our experiments, the contiguity of sequences can affect the optimal choice of τ . For datasets with larger N50, the similarity of original features between sequence fragments generated through data augmentation is lower, and they are more suitable for smaller τ values. The smaller τ increases the value of $\exp(\cos(z_{i,k}, z_{i,k'})/\tau)$, causing the loss function to become steeper for similar instances. The model then concentrates more on bringing the representations of similar instances (the sub-sequences generated from the same original contig) closer.

Comment — Line 412: What are the early stopping criteria?

Response: After ten epochs, training is stopped if the accuracy of instance discrimination (the self-supervised learning task) consistently exceeds 0.99 for three consecutive epochs.

Comment — Line 424: “on community division algorithm Leiden” => “on the Leiden community division algorithm”

Response: We have modified it accordingly.

Comment — Line 441: It could be helpful to explicitly state here that you used tetra-nucleotide frequencies (TNF).

Response: We have revised the original title to “Tetra-nucleotide Frequencies (TNF)”.

Comment — Line 449: Why would zero vectors occur? The vector represents all possible 4-mers, while treating reverse compliments as equivalent.

Response: Apologies for any confusion. We intended to convey that there are elements with zero values in the vector. We have modified it accordingly.

Comment — Line 454: How were reads mapped to the contigs? What filtering of the raw read mapping was conducted (e.g., removing improper read pairs)? What software (and parameters) were used to obtain per-position (bp) coverage information?

Response: The reads were aligned to the contigs using BWA (version 0.7.17) [22], and the per-position coverage information for the contigs was calculated using BEDTools (version 2.30.0). We have now included these details in the section “Compared methods and experimental settings” for clarity (see L633-L635). The Water Group datasets used in this study comprise clean reads (Q20 reads; reads with an average quality > 20) provided by the researchers [16]. We did not conduct any filtering to the reads as done in the papers that utilized the datasets for binning benchmarking (STEC dataset [23] and MetaHIT dataset [1, 10]).

In this response letter, we have additionally filtered the reads through read trimming, and removal of human reads using the MetaWRAP “Read_QC” module [24]. In MetaWRAP “Read_QC” module, read trimming was accomplished using Trim Galore (<https://github.com/FelixKrueger/TrimGalore>) with default parameters. We subsequently assembled the filtered reads using MEGAHIT [25]. As shown in Figure R2, COMEBin achieves the best performance in terms of the number of recovered near-complete bins (>90% completeness and <5% contamination). Compared with the best of other methods, COMEBin has an average improvement of 25.7% on the three datasets. The average improvement matches the improvement (28.0%) of the original datasets.

Comment — Line 454: Is COMEBin suitable for single-end data? I’m assuming that it is not, but this should be explicitly stated.

Response: The input to COMEBin consists of contigs and BAM files (generated by mapping reads to the contigs). It is not restricted to specific sequencing data types.

Comment — Line 519: “got” (and “get”) is too colloquial. Please re-word.

Response: We have modified it into “obtained” accordingly.

Comment — Line 522: Explicitly state how edges were filtered/retained.

Response: We have stated it as follows:

“To focus on edges with low distances, we kept 50%, 80%, or 100% of edges with smaller values for subsequent clustering.”

Comment — Line 547: What parameters were assessed? Which optimization method was used (e.g., grid search, random parameter search, bayesian optimization, etc)?

Response: Different parameters include σ in Equation 13, resolution parameters, and edge ratios (proportions of edges kept for clustering). And we have added related descriptions in the section “Clustering” (see L527-L530).

We used grid search, and the candidate parameter value ranges were as follows:

σ in Equation 13: [0.05, 0.1, 0.15, 0.2, 0.3]

Resolution parameters in Leiden: [1, 5, 10, 30, 50, 70, 90, 110]

Edge ratios (proportions of edges kept for clustering): [50%, 80%, 100%]

The final results were determined using the method described in the “Choose the best result automatically” section.

Comment — Figures 1a, 2a, and 3a: Reduce the plots to just “comp>90% and cont<5” and “comp>50% and cont<5”. The other plots can go in the supplemental materials.

Response: We have represented the number of bins using stacked bar plots in Fig 1.a, Fig 2.a, and Fig 3.a. We have kept bins with contamination <5% and varying completeness thresholds, considering both reviewers’ comments regarding Figures 1-3.

Comment — Figures 1b and 2b: Plot by accuracy measure instead of by dataset (e.g., color by dataset and facet by accuracy measure). Show the actual accuracy values instead of the sum-of-ranks.

Response: We have presented the actual accuracy values instead of the sum-of-ranks in Fig 1.a, Fig 2.a,

Comment — Figure 5: “contig and” => “contig, and”. Which ordination method was used? There is over-plotting of points, which can hide some inaccurate clustering. It can help to reduce the alpha (transparency) of the point colors in order to better visualize overlap.

Response: The t-SNE [26] was used for the ordination and visualization of the data, and we have added the related descriptions in the context. We have adjusted the transparency of the point colors.

Comment — Figure S2: Please define “known” and “unknown” for the context of this figure.

Response: “Known” genomes refer to bins that can be annotated at the species level using GTDB-Tk, and “Unknown” otherwise. We have added the related descriptions in the context.

Comment — Table S1: “(10- sample)” is formatted oddly. Explicitly define ”best results” in the table caption.

Response: We have replaced this table with Supplementary Fig. S6, and the format of “(10- sample)” has been correspondingly resolved.

Comment — Table S3: Briefly summarize the ablation experiment in the table caption. Explicitly define “best results”.

Response: We have replaced this table with Supplementary Fig. S7 and added the descriptions of the experiments accordingly.

Comment — Table S4: Briefly describe “views”. Explicitly define “best results”.

Response: We have replaced this table with Supplementary Fig. S8, and added the descriptions of “views” correspondingly.

Comment — Table S5: Is “minutes” in wall-clock time or CPU minutes? Given that resource profiling is dependent on other processing occurring on the machine(s) at the time of profiling, multiple replicates should be conducted, with the aggregate and variance among replicates reported.

Response: The “minutes” is in wall-clock time. We have run each tool on each dataset three times and calculated the average running time and memory usage. The relevant results are shown in Supplementary Table S1.

Comment — Table S6: “Dataset” => “Datasets”. Include the raw read lengths, read quality, sequencing platform, and library preparation method (if available).

Response: We have modified “Dataset” to “Datasets” and added raw read lengths, read quality (Q20 (%)), and sequencing platform into Table S7. The term “Q20 (%)” represents the fraction of reads with an average quality > 20.

Comment — Algorithm S1: “(com” => “(com)”. “(cov” => ”(cov)”

Response: We have modified it.

Comment — Section S2.1: SemiBin can be used to bin contigs as done by VAMB and CLMB. See <https://github.com/BigDataBiology/SemiBin#easy-multi-samples-binning-mode>. It would be best to re-evaluate SemiBin using this “combined-contigs” method, as done with VAMB and CLMB.

Response: Our descriptions in the main text may have caused some misunderstanding. We didn’t mean that SemiBin can not be used in multi-sample mode. There are two available ways for this mode. VAMB [10] and CLMB [11] concatenate sample-specific contigs from all the samples for binning and split the bins by sample information. SemiBin’s multi-sample mode differs slightly from VAMB’s. As stated in the paper of SemiBin, SemiBin [5] uses the sample-specific contigs for binning, and abundance information is aggregated across samples. This approach can be easily adopted by other bidders without modifying the codes. Therefore, we adopted this approach and evaluated all binning methods, excluding VAMB and CLMB. While SemiBin’s “easy-multi-samples-binning-mode” concatenates all the sample-specific contigs for ease of operation, SemiBin bins the sample-specific contigs generated from each sample individually.

Fig. R1 Comparison of COMEBin using different output dimensions for the “Coverage network”.

Fig. R2 Performance of the binning methods on the datasets with filtered reads. We additionally filtered the reads through read trimming and removal of human reads using the MetaWRAP “Read_QC” module [24]. In MetaWRAP “Read_QC” module, read trimming was accomplished using Trim Galore (<https://github.com/FelixKrueger/TrimGalore>) with default parameters. We subsequently assembled the filtered reads using MEGAHIT [25].

References

- [1] Kang, D.D., Froula, J., Egan, R., Wang, Z.: MetaBAT, an efficient tool for accurately reconstructing single genomes from complex microbial communities. *PeerJ* **3**, 1165 (2015)
- [2] Sczyrba, A., Hofmann, P., Belmann, P., Koslicki, D., Janssen, S., Dröge, J., Gregor, I., Majda, S., Fiedler, J., Dahms, E., *et al.*: Critical Assessment of Metagenome Interpretation—a benchmark of metagenomics software. *Nature Methods* **14**(11), 1063–1071 (2017)
- [3] Meyer, F., Fritz, A., Deng, Z.-L., Koslicki, D., Lesker, T.R., *et al.*: Critical assessment of metagenome interpretation: the second round of challenges. *Nature Methods* **19**(4), 429–440 (2022)
- [4] Meyer, F., Lesker, T.R., Koslicki, D., Fritz, A., Gurevich, A., Darling, A.E., Sczyrba, A., Bremges, A., McHardy, A.C.: Tutorial: assessing metagenomics software with the CAMI benchmarking toolkit. *Nature Protocols* **16**(4), 1785–1801 (2021)
- [5] Pan, S., Zhu, C., Zhao, X.M., Coelho, L.P.: A deep siamese neural network improves metagenome-assembled genomes in microbiome datasets across different environments. *Nature Communications* **13**(1), 2326 (2022)
- [6] Liu, C.-C., Dong, S.-S., Chen, J.-B., Wang, C., Ning, P., Guo, Y., Yang, T.-L.: Metadecoder: a novel method for clustering metagenomic contigs. *Microbiome* **10**(1), 1–16 (2022)
- [7] Chklovski, A., Parks, D.H., Woodcroft, B.J., Tyson, G.W.: CheckM2: a rapid, scalable and accurate tool for assessing microbial genome quality using machine learning. *Nature Methods* **20**(8), 1203–1212 (2023)
- [8] Pan, S., Zhao, X.M., Coelho, L.P.: SemiBin2: self-supervised contrastive learning leads to better MAGs for short- and long-read sequencing. *Bioinformatics* **39**(39 Suppl 1), 21–29 (2023)
- [9] Arisdakessian, C.G., Nigro, O.D., Steward, G.F., Poisson, G., Belcaid, M.: CoCoNet: an efficient deep learning tool for viral metagenome binning. *Bioinformatics* **37**(18), 2803–2810 (2021)
- [10] Nissen, J.N., Johansen, J., Allesøe, R.L., Sønderby, C.K., Armenteros, J.J.A., *et al.*: Improved metagenome binning and assembly using deep variational autoencoders. *Nature Biotechnology* **39**(5), 555–560 (2021)
- [11] Zhang, P., Jiang, Z., Wang, Y., Li, Y.: CLMB: deep contrastive learning for robust metagenomic binning. In: *Research in Computational Molecular Biology: 26th Annual International Conference, RECOMB 2022, San Diego, CA, USA, May 22–25, 2022, Proceedings*, pp. 326–348 (2022). Springer
- [12] Nayfach, S., Roux, S., Seshadri, R., Udwy, D., Varghese, N., Schulz, F., Wu, D., Paez-Espino, D., Chen, I.M., Huntemann, M., Palaniappan, K., Ladau, J., Mukherjee, S., Reddy, T.B.K., Nielsen, T., Kirton, E., Faria, J.P., Edirisinghe, J.N., Henry, C.S., Jungbluth, S.P., Chivian, D., Dehal, P., Wood-Charlson, E.M., Arkin, A.P., Tringe, S.G., Visel, A., Woyke, T., Mouncey, N.J., Ivanova, N.N., Kyrpides, N.C., Eloe-Fadrosh, E.A., *et al.*: A genomic catalog of Earth’s microbiomes. *Nature Biotechnology* **39**(4), 499–509 (2021)
- [13] Wang, Y., Leung, H.C., Yiu, S.M., Chin, F.Y.: MetaCluster 4.0: a novel binning algorithm for NGS reads and huge number of species. *J Comput Biol* **19**(2), 241–249 (2012)
- [14] Kang, D.D., Li, F., Kirton, E., Thomas, A., Egan, R., An, H., Wang, Z.: MetaBAT 2: an adaptive binning algorithm for robust and efficient genome reconstruction from metagenome assemblies. *PeerJ*

7, 7359 (2019)

- [15] Jaiswal, A., Babu, A.R., Zadeh, M.Z., Banerjee, D., Makedon, F.: A survey on contrastive self-supervised learning. *Technologies* **9**(1), 2 (2020)
- [16] Liang, J., Mao, G., Yin, X., Ma, L., Liu, L., Bai, Y., Zhang, T., Qu, J.: Identification and quantification of bacterial genomes carrying antibiotic resistance genes and virulence factor genes for aquatic microbiological risk assessment. *Water Research* **168**, 115160 (2020)
- [17] Meyer, F., Hofmann, P., Belmann, P., Garrido-Oter, R., Fritz, A., Sczyrba, A., McHardy, A.C.: AMBER: Assessment of Metagenome BinnERs. *Gigascience* **7**(6), 069 (2018)
- [18] de Nies, L., Lopes, S., Busi, S.B., Galata, V., Heintz-Buschart, A., Laczny, C.C., May, P., Wilmes, P.: PathoFact: a pipeline for the prediction of virulence factors and antimicrobial resistance genes in metagenomic data. *Microbiome* **9**(1), 49 (2021)
- [19] Alcock, B.P., Huynh, W., Chalil, R., Smith, K.W., Raphenya, A.R., *et al.*: and resistome prediction at the Comprehensive Antibiotic Resistance Database. *Nucleic Acids Research* **51**(D1), 690–699 (2023)
- [20] Altschul, S.F., Gish, W., Miller, W., Myers, E.W., Lipman, D.J.: Basic local alignment search tool. *Journal of molecular biology* **215**(3), 403–410 (1990)
- [21] Blin, K., Shaw, S., Kloosterman, A.M., Charlop-Powers, Z., van Wezel, G.P., Medema, M.H., Weber, T.: antiSMASH 6.0: improving cluster detection and comparison capabilities. *Nucleic Acids Research* **49**(W1), 29–35 (2021)
- [22] Li, H.: Aligning sequence reads, clone sequences and assembly contigs with bwa-mem. arXiv preprint arXiv:1303.3997 (2013)
- [23] Ma, T., Xiao, D., Xing, X.: MetaBMF: a scalable binning algorithm for large-scale reference-free metagenomic studies. *Bioinformatics* **36**(2), 356–363 (2020)
- [24] Uritskiy, G.V., DiRuggiero, J., Taylor, J.: MetaWRAP-a flexible pipeline for genome-resolved metagenomic data analysis. *Microbiome* **6**(1), 158 (2018)
- [25] Li, D., Liu, C.M., Luo, R., Sadakane, K., Lam, T.W.: MEGAHIT: an ultra-fast single-node solution for large and complex metagenomics assembly via succinct de Bruijn graph. *Bioinformatics* **31**(10), 1674–1676 (2015)
- [26] Van der Maaten, L., Hinton, G.: Visualizing data using t-sne. *Journal of Machine Learning Research* **9**(11), 2579–2605 (2008)

REVIEWERS' COMMENTS

Reviewer #1 (Remarks to the Author):

I thank the authors for their answers and changes made to manuscript that overall address my previous concerns.

I think the paper is ready for publication.

Below are a few minor comments:

L106 – L109 ‘the simulated datasets’

Remove “the”

L146 : The presence of common strains

Replace common by ‘similar’ or ‘closely related’

L184: “on the species level”

Replace by ‘at the species level’

Reviewer #2 (Remarks to the Author):

The authors have adequately addressed all of my comments. I commend the authors on their great work!

Response Letter

Dear Editors and Reviewers,

We greatly appreciate your time and efforts in the manuscript. We thank the reviewers' positive comments. In the revision, we have further modified our manuscript accordingly. In the following, we present one-by-one responses to the first reviewer's comments.

Reviewer #1:

Comment — I thank the authors for their answers and changes made to the manuscript that overall address my previous concerns. I think the paper is ready for publication.

Response: Thank you for the comments.

Comment — Below are a few minor comments:

L106 – L109 “the simulated datasets” Remove “the”

L146 : The presence of common strains: Replace common by “similar” or “closely related”

L184: “on the species level”: Replace by “at the species level”

Response: We have made the modifications according to the comments.